# Transmission networks of SARS-CoV-2 in Coastal Kenya during the first two waves: A retrospective genomic study

Charles N Agoti[1,2]*, Lynette Isabella Ochola-Oyier[1], Simon Dellicour[3,4], Khadija Said Mohammed[1], Arnold W Lambisia[1], Zaydah R de Laurent[1], John M Morobe[1], Maureen W Mburu[1], Donwilliams O Omuoyo[1], Edidah M Ongera[1], Leonard Ndwiga[1], Eric Maitha[5], Benson Kitole[5], Thani Suleiman[5], Mohamed Mwakinangu[5], John K Nyambu[5], John Otieno[5], Barke Salim[5], Jennifer Musyoki[1], Nickson Murunga[1], Edward Otieno[1], John N Kiiru[5], Kadondi Kasera[5], Patrick Amoth[5], Mercy Mwangangi[5], Rashid Aman[5], Samson Kinyanjui[1,2,6], George Warimwe[1,6], My Phan[7], Ambrose Agweyu[1], Matthew Cotten[7,8], Edwine Barasa[1], Benjamin Tsofa[1], D James Nokes[1,9], Philip Bejon[1,6], George Githinji[1,2]

[1]Kenya Medical Research Institute (KEMRI)-Wellcome Trust Research Programme, Kilifi, Kenya; [2]Pwani University, Kilifi, Kenya; [3]Spatial Epidemiology Lab (SpELL), Université Libre de Bruxelles, Bruxelles, Belgium; [4]Department of Microbiology, Immunology and Transplantation, Rega Institute, Laboratory for Clinical and Epidemiological Virology, KU Leuven, University of Leuven, Leuven, Belgium; [5]Ministry of Health, Nairobi, Kenya; [6]Nuffield Department of Medicine, University of Oxford, Oxford, United Kingdom; [7]Medical Research Centre (MRC)/ Uganda Virus Research Institute, Entebbe, Uganda; [8]MRC-University of Glasgow Centre for Virus Research, Glasgow, United Kingdom; [9]University of Warwick, Coventry, United Kingdom

*For correspondence: cnyaigoti@kemri-wellcome.org

Competing interest: The authors declare that no competing interests exist.

## Abstract

**Background:** Detailed understanding of severe acute respiratory syndrome coronavirus 2 (SARS-CoV-2) regional transmission networks within sub-Saharan Africa is key for guiding local public health interventions against the pandemic.

**Methods:** Here, we analysed 1139 SARS-CoV-2 genomes from positive samples collected between March 2020 and February 2021 across six counties of Coastal Kenya (Mombasa, Kilifi, Taita Taveta, Kwale, Tana River, and Lamu) to infer virus introductions and local transmission patterns during the first two waves of infections. Virus importations were inferred using ancestral state reconstruction, and virus dispersal between counties was estimated using discrete phylogeographic analysis.

**Results:** During Wave 1, 23 distinct Pango lineages were detected across the six counties, while during Wave 2, 29 lineages were detected; 9 of which occurred in both waves and 4 seemed to be Kenya specific (B.1.530, B.1.549, B.1.596.1, and N.8). Most of the sequenced infections belonged to lineage B.1 (n = 723, 63%), which predominated in both Wave 1 (73%, followed by lineages N.8 [6%] and B.1.1 [6%]) and Wave 2 (56%, followed by lineages B.1.549 [21%] and B.1.530 [5%]). Over the study period, we estimated 280 SARS-CoV-2 virus importations into Coastal Kenya. Mombasa City, a vital tourist and commercial centre for the region, was a major route for virus imports, most of which occurred during Wave 1, when many Coronavirus Disease 2019 (COVID-19) government restrictions were still in force. In Wave 2, inter-county transmission predominated, resulting in the emergence of local transmission chains and diversity.

**Conclusions:** Our analysis supports moving COVID-19 control strategies in the region from a focus on international travel to strategies that will reduce local transmission.

**Funding:** This work was funded by The Wellcome (grant numbers: 220985, 203077/Z/16/Z, 220977/Z/20/Z, and 222574/Z/21/Z) and the National Institute for Health and Care Research (NIHR), project references: 17/63/and 16/136/33 using UK Aid from the UK government to support global health research, The UK Foreign, Commonwealth and Development Office. The views expressed in this publication are those of the author(s) and not necessarily those of the funding agencies.

## Editor's evaluation

It is important to describe patterns of SARS-CoV-2 spread across the globe, beyond high-income countries. This study provides and evaluates SARS-CoV-2 sequence data from ~1200 PCR confirmed COVID-19 patients in Coastal Kenya to characterize phylogenetically likely importation and geographic infection routes, as well as the emergence of geographically distinct SARS-CoV-2 lineages.

## Introduction

Coronavirus Disease 2019 (COVID-19), caused by the severe acute respiratory syndrome coronavirus 2 (SARS-CoV-2), was declared a pandemic on March 11, 2020 (*Hu et al., 2021*). By February 28, 2021, there had been at least 114 million confirmed cases of COVID-19 and more than 2.6 million deaths worldwide (https://covid19.who.int/). By the same date, Kenya, an East Africa country with a population of around 50 million people, had reported a total of 105,648 COVID-19 cases and 1856 associated deaths, most of which were associated with two distinct waves of infections (*MOH, 2021*).

Kenya reported its first COVID-19 case on March 13, 2020. In response, the government outlined a series of countermeasures to minimize the effects of a pandemic locally (*Brand et al., 2021*). For instance, international travel was restricted, international borders closed, public gatherings prohibited, meetings with over 15 participants forbidden, travel from hotspot counties restricted, places of worship, bars, schools, and other learning institutions closed, and a nationwide dusk-to-dawn curfew enforced (*Wambua et al., 2022*). Despite these measures, the COVID-19 case numbers consistently grew and serological surveys in June 2020 indicated the local epidemic had progressed more than it could be discerned from the limited laboratory testing (*Etyang et al., 2021*; *Uyoga et al., 2021a*).

An analysis of blood donor samples collected in the first quarter of 2021 found that anti-SARS-CoV-2 IgG prevalence in Kenya was 48.5% (*Adetifa et al., 2021*; *Uyoga et al., 2021b*). Despite this progression of the local epidemic, understanding of local SARS-CoV-2 spread patterns remains limited (*Githinji et al., 2021*; *Wilkinson et al., 2021*). During the first two waves, documented cases were concentrated in the major cities, with Nairobi, the capital, accounting for a cumulative total of ~42% of the cases by February 2021 and Mombasa, a coastal city, accounting for ~8% of the cases (*Brand et al., 2021*). Here, we focused on the latter and its environs.

Throughout the COVID-19 pandemic period, genomic analysis has been crucial for tracking the spread of SARS-CoV-2 and investigating its transmission pathways (*Bugembe et al., 2020*; *Geoghegan et al., 2020*; *Oude Munnink et al., 2020*; *Worobey et al., 2020*). Previously, we analysed 311 SARS-CoV-2 early genomes collected in Coastal Kenya during Wave 1 (*Githinji et al., 2021*). In that study, we showed that several Pango lineages had been introduced into Coastal Kenya, but most of them did not take off, except for lineage B.1 (*Githinji et al., 2021*).

The second SARS-CoV-2 wave of infections in Kenya began in mid-September 2020 (*Figure 1A*), and a mathematical modelling study suggested that this wave was primarily driven by the easing of government restrictions (*Brand et al., 2021*). Here, we utilized a large set of genome sequences from Coastal Kenya to rule out that a new more transmissible or immune evasive variant was not involved in the second wave and investigate patterns of virus importations, lineage temporal dynamics, and local spread patterns within and between the six counties of Coastal Kenya during the first two epidemic waves of SARS-CoV-2 infections in Kenya.

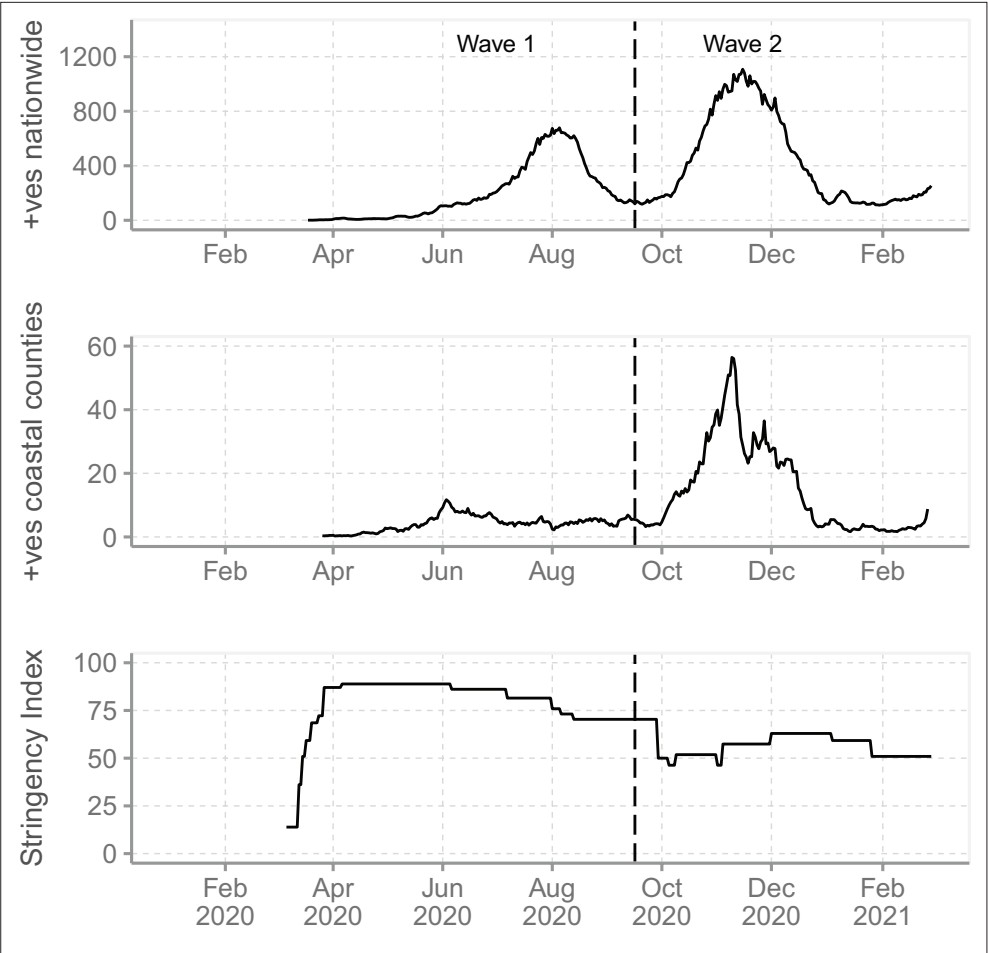

**Figure 1.** The severe acute respiratory syndrome coronavirus 2 (SARS-CoV-2) epidemic in Kenya and government response. (**A**) The reported daily new cases in Kenya from March 2020 to February 2021 shown as 7-day-rolling average demonstrating the first two national SARS-CoV-2 waves of infections. (**B**) The total reported daily cases for Coastal Kenya counties during the study period shown as 7-day-rolling average per million people. (**C**) The Kenya government COVID-19 intervention level during the study period as summarized by the Oxford Stringency Index (SI) (*Hale et al., 2021*).

The online version of this article includes the following source data for figure 1:

**Source data 1.** Number of daily new cases of severe acute respiratory syndrome coronavirus 2 (SARS-CoV-2) in Kenya up to February 26, 2021, and the corresponding 7-day-rolling average.

**Source data 2.** Number of daily positive tests per million people for the Coastal Kenya region (all six counties combined).

**Source data 3.** Kenya government Coronavirus Disease 2019 (COVID-19) restrictions stringency index during the study period.

## Methods
### Study design and population

We analysed SARS-CoV-2 genomic sequences from nasopharyngeal/oropharyngeal (NP/OP) swab samples collected across the six coastal counties of Kenya (Mombasa, Kilifi, Kwale, Taita Taveta, Tana River, and Lamu) between March 17, 2020, and February 26, 2021. Of the six, Mombasa is the most densely populated and has a seaport, an international airport, and an island (*Table 1*). Kwale and Taita Taveta counties share a border with Tanzania while Lamu includes several islands in the Indian Ocean. Based on the observed nationwide peaks in SARS-CoV-2 infections, we divided the study period into (a) Wave 1, which was the period between March 17 and September 15, 2020, and (b) Wave 2, the period between September 16, 2020, and February

**Table 1.** Number of severe acute respiratory syndrome coronavirus 2 (SARS-CoV-2) positives reported by the Ministry of Health in Kenya by February 26, 2021, and breakdown of those conducted at KEMRI-Wellcome Trust Research Programme (KWTRP), including status of sequencing.

| County | Total Population size[*] (%) | Population density[†] | Ministry of Health reported positves [‡] (%) | RT-PCR tests (KWTRP, %) | Positives (KWTRP, %) | No. of whole genomes sequenced (%)[§] |
|---|---|---|---|---|---|---|
| Mombasa | 1,208,333 (27.9) | 5,495 | 8450 (66.8) | 46,143 (55.8) | 3139 (49.6) | 468 (41.1) |
| Kilifi | 1,453,787 (33.6) | 116 | 2458 (19.4) | 12,908 (15.6) | 1443 (22.8) | 294 (25.8) |
| Kwale | 866,820 (20.0) | 105 | 436 (3.4) | 5491 (6.6) | 436 (6.9) | 102 (9.0) |
| Taita Taveta | 340,671 (7.9) | 20 | 855 (6.7) | 14,543 (17.6) | 855 (13.5) | 196 (13.5) |
| Tana River | 315,943 (7.3) | 8 | 106 (0.8) | 877 (1.1) | 106 (1.7) | 16 (1.7) |
| Lamu | 143,920 (3.3) | 23 | 350 (2.7) | 2754 (3.3) | 350 (5.5) | 63 (5.5) |
| Overall | 4,329,474 (100.0) | 52 | 12,655 (100.0) | 82,716 (100.0) | 6329 (100.0) | 1139 (100.0) |

[*]Number of residents as per the 2019 national population census.

[†]Units here are number of persons per square kilometre.

[‡]The Ministry of Health reports compiled results from all testing centres across the country including KWTRP.

[§]The numbers in brackets represents the proportion sequenced of those detected following RT-PCR at the KWTRP.

26, 2021 (*Figure 1A and B*). Wave 2 period began when the number of national daily positive cases started to show a renewed consistent rise after the peak of Wave 1.

## Ethical statement

The study protocol was reviewed and approved by the Scientific and Ethics Review Committee (SERU) at Kenya Medical Research Institute (KEMRI), Nairobi, Kenya (SERU protocol #4035). The committee did not require individual patient consent for studies using residual diagnostic material to investigate the SARS-CoV-2 genomic epidemiology for improved public health response.

## Samples analysed

The study used residue NP/OP swab samples collected by the Ministry of Health (MoH) County Department of Health rapid response teams (RRTs) for SARS-CoV-2 diagnostic testing (*Agoti et al., 2020*; *Nyagwange et al., 2022*). The RRTs delivered the NP/OP swabs to the KEMRI-Wellcome Trust Research Programme (KWTRP) laboratories within 48 hr in cool boxes with ice packs. The samples were from persons of any age collected following the MoH eligibility criteria that were periodically revised. Participants included persons with (1) acute respiratory illness symptoms, (2) returning travellers from early COVID-19 hotspot countries (i.e. China, Italy, and Iran), (3) persons seeking entry into Kenya at international border points, (4) contacts of confirmed cases, and (5) persons randomly approached as part of the 'mass' testing effort to understand the extent of infection spread in the communities.

## SARS-CoV-2 testing and genome sequencing at KWTRP

To purify nucleic acids (NA) in the NP/OP samples, a variety of commercial kits were used, namely, QIAamp Viral RNA Mini Kit, RNeasy QIAcube HT Kit, QIASYMPHONY RNA Kit, TIANamp Virus RNA Kit, Da An Gene Nucleic acid Isolation and Purification Kit, SPIN X Extraction, and RADI COVID-19 detection Kit. The NA extracts were tested for SARS-CoV-2 genetic material using one of the following kits/protocols: (1) the Berlin (Charité) primer-probe set (targeting envelope [E] gene, nucleocapsid [N] or RNA-dependent RNA-polymerase [RdRp]), (2) European Virus Archive – GLOBAL (EVA-g) (targeting E or RdRp genes), (3) Da An Gene Co. detection Kit (targeting N or ORF1ab), (4) BGI RT-PCR kit (targeting ORF1ab), (5) Sansure Biotech Novel Coronavirus (2019-nCoV) Nucleic Acid Diagnostic real-time RT-PCR kit or (6) Standard M kit (targeting E and ORF1ab), and (7) TIB MOLBIOL kit (targeting E gene). Kit/protocol-determined cycle threshold cut-offs were used to define positives (*Mohammed et al., 2020*).

Though we initially intended to sequence every positive case diagnosed at KWTRP, eventually we settled on sequencing a subset of cases once the epidemic had established (*Githinji et al., 2021*). Samples sequenced were those with RT-PCR cycle threshold values of <30 with spatial (at county level) and temporal (by month) representation (*Figure 2—figure supplement 1*). We re-extracted

NA from samples selected for sequencing using QIAamp Viral RNA Mini kit following the manufacturer's instructions and reverse-transcribed the RNA using LunaScript RT SuperMix Kit. The cDNA was amplified using Q5 Hot Start High-Fidelity 2x Mastermix along with the ARTIC nCoV-2019 version 3 primers. The PCR products were run on a 1.5% agarose gel, and for samples whose SARS-CoV-2 amplification was considered successful (amplicons visible) were purified using Agencourt AMPure XP beads and taken forward for library preparation. Sequencing libraries were constructed using Oxford Nanopore Technologies (ONT) ligation sequencing kit and the ONT Native Barcoding Expansion kit as described in the ARTIC protocol (*Tyson et al., 2020*). Every MinION (Mk1B) run comprised 23 samples and 1 negative (no-template) control.

## Genome assembly and lineage assignment

Following MinION sequencing, the FAST5 files were base-called and demultiplexed using the ONT's software Guppy v3.5–4.2. Consensus SARS-CoV-2 sequences were derived from the reads using the ARTIC bioinformatics pipeline (https://artic.network/ncov-2019/ncov2019-bioinformatics-sop.html; last accessed August 3, 2021). A threshold of ×20 read depth was required for a base to be included in the consensus genome; otherwise, it was masked with an N (*Githinji et al., 2021*). Only complete or near-complete genomes with N count <5980 (i.e. >80% coverage) were further analysed.

The consensus genomes were assigned into Pango lineages as described by *Rambaut et al., 2020* using Pangolin v3.1.16 (command line version) with Pango v1.2.101 and PangoLEARN model v2021-11-25 (*O'Toole et al., 2021*). Contextual information about lineages was obtained from the Pango lineage description list available at https://cov-lineages.org/lineage_list.html (last accessed December 21, 2021). Variants of concern (VOC) and variants of interest (VOI) were designated based on the WHO framework as of May 31, 2021 (https://www.who.int/en/activities/tracking-SARS-CoV-2-variants/). Amino acid sequence changes in the Coastal Kenya genomes were investigated using the Nextclade tool v0.14.2 (*Hadfield et al., 2018*): https://clades.nextstrain.org/ (last accessed August 3, 2021). Mutations in the Kenyan lineages were visualized using the Stanford University CORONAVIRUS ANTIVIRAL & RESISTANCE Database tool on webpage: https://covdb.stanford.edu/page/mutation-viewer/ (last accessed August 3, 2021).

## Global contextual sequences

The global contextual sequences were obtained from GISAID (https://www.gisaid.org/) using the inclusion criteria: (1) presence of the full sample collection date (year–month–day), (2) host recorded as 'Human', (3) sample collected between March 1, 2020, and February 28, 2021, and (4) absence of >5980 ambiguous (N) nucleotides. Three analysis datasets were prepared as shown in *Figure 5—figure supplement 1*.

a.  Set 1 was for investigating the global context and temporal dynamics of the Pango lineages detected in Coastal Kenya. All data available on GISAID assigned Pango lineages detected in Coastal Kenya were included (n = 420,492).
b.  Set 2 was for investigating lineage temporal dynamics across widening scales of observation (Coastal Kenya, across Kenya, Eastern Africa, Africa, and globally). These included all eligible African genomes (n = 21,150) and a subset of non-African genomes selected randomly from 'master dataset' using the R randomization command: *sample_n()*. A maximum of 30 genomes were selected from each country by year and month. The Eastern Africa subset comprised of 5275 genomes from 10 countries, namely, Ethiopia, Uganda, Rwanda, Malawi, Zimbabwe, Zambia, Mozambique, Madagascar, Reunion (a France overseas territory), and The Comoros.
c.  Set 3 was for investigating global phylogenetic relationships. It included genomes from the global subset of lineages detected in Coastal Kenya and then randomly split into two subsamples for tractable subsequent phylogenetic analysis (*Figure 5—figure supplement 1*).

## Phylogenetic analysis

Multiple sequence alignments were prepared in Nextalign v0.1.6 software using the initial Wuhan sequence (Accession number: NC_045512) as the reference with the command:

$$nextalign - r\,NC\_045512.fasta - i\,input.fasta$$

The alignment was manually inspected in AliView v1.21 to spot any obvious problems/misalignments. Quick non-bootstrapped neighbour-joining trees were created in SEAVIEW v4.6.4 to identify any aberrant sequences which were henceforth discarded. Maximum likelihood (ML) phylogenies were reconstructed using IQTREE v2.1.3 under the GTR (general time-reversible) model of evolution using the command:

$$./iqtree2 - s\,input.aligned.fasta - nt\,4 - m\,GTR$$

The ML tree was linked to the various metadata (lineage, county, source, etc.) in R programming software v4.0.2 and visualized using the R package 'ggtree' v2.4.2. The ML phylogenetic tree was subsequently time-calibrated with the program TreeTime, assuming a constant genomic evolutionary rate of SARS-CoV-2 of $8.4 \times 10^{-4}$ nucleotide substitutions per site per year (*Sagulenko et al., 2018*), and using the command.

$$treetime\ -\ -tre\,input.aligned.fasta.treefile\ -\ -aln\,input.aligned.fasta\ -\ -clock\ -\ rate\,0.00084\ -$$
$$dates\,dates.csv$$

Outlier sequences deviating from the molecular clock were identified by TreeTime and excluded using the R package 'treeio'. TempEst v1.5.3 was then used to assess the consistency of nucleotide evolution of the analysed data with a molecular clock. A linear regression of root-to-tip genetic distances against sampling dates was plotted in RStudio and the coefficient of determination ($R^2$) assessed. The resulting trees were visualized using the R package 'ggtree' v2.4.2.

## Import/export analysis

We estimated the number of viral importation/exportation events between Coastal Kenya and the rest of the world by ancestral state reconstruction from the global ML tree using methods similar to those described by *Tegally et al., 2021*; *Wilkinson et al., 2021*. This was achieved using the date and location annotated tree topology to count the number of transitions between Coastal Kenya counties and the rest of the world ('non-coastal Kenya') using the Python script developed by the KwaZuluNatal Research Innovation & Sequencing Platform team (KRISP, https://github.com/krisp-kwazu-lu-natal/SARSCoV2_South_Africa_major_lineages/tree/main/Phylogenetics; last accessed August 4, 2021). The results were plotted in R using the package 'ggplot2' v3.3.3. This analysis was repeated with a further two subsamples of the global background data and with also a downsampled set of the Coastal Kenya genomes that were normalized spatially and temporally (*Supplementary file 5*).

## Phylogeographic analyses

We used a discrete phylogeographic approach (*Lemey et al., 2009*) to investigate the dispersal history of SARS-CoV-2 lineages among coastal counties while trying to mitigate the potential impact of sampling bias by subsampling Kenyan counties according to their relative epidemiological importance during the study period. For this purpose, we implemented a subsampling procedure similar to the one described by Dellicour and colleagues to analyse the circulation of SARS-CoV-2 among New York City boroughs during the first phase of the American epidemic (*Dellicour et al., 2021b*). Specifically, we performed replicated discrete phylogeographic analyses based on random subset of genomic sequences. Each subset was obtained by subsampling available Kenyan genomic sequences according to the COVID-19 incidence recorded in each sampled county during the study period (Mombasa: 699 cases/100,000 people; Kilifi: 169; Kwale: 50; Taita Taveta: 251; Tana River: 34; and Lamu: 243; *Table 1*). Because Lamu was the proportionally least sampled county when comparing available number of sequences to local incidence, the sampling intensity of this county (63 genomic sequences sampled for a recorded incidence of 243 cases per 100,000 people) served as reference for downsampling the available number of sequences from the other counties. The resulting downsampled data sets comprised the following number of sequences: n = 181 (Mombasa), 44 (Kilifi), 13 (Kwale), 65 (Taita Taveta), 9 (Tana River), and 63 (Lamu). To investigate the impact of the stochastic subsampling procedure, we performed 10 replicated analyses each based on a distinct subsampling.

Discrete phylogeographic inferences were all performed using the discrete diffusion model (*Lemey et al., 2009*) implemented in the software package BEAST 1.10 (*Suchard et al., 2018*). In a first

time and following a previously described analytical pipeline (*Dellicour et al., 2021a*), a preliminary discrete phylogeographic reconstruction was performed to delineate clades corresponding to distinct introduction events of SARS-CoV-2 lineages into Kenya. For this initial phylogeographic analysis, we only considered two possible ancestral locations: 'Kenya' and 'other location'. We conducted Bayesian inference through Markov chain Monte Carlo (MCMC) for $10^6$ iterations and sampled every $10^3$ iterations. To ensure that effective sample size (ESS) values associated with estimated parameters were all >200, we inspected MCMC convergence and mixing properties using the program Tracer 1.7 (*Rambaut et al., 2018*). We then generated a maximum clade credibility (MCC) tree using the program TreeAnnotator 1.10 (*Suchard et al., 2018*) after having discarded 10% of sampled trees as burn-in. Finally, we used the resulting MCC tree to delineate phylogenetic clades corresponding to independent introduction events into Kenya.

In a second time, each replicated phylogeographic analysis was conducted along the overall time-scaled phylogenetic tree previously obtained with TreeTime (see the 'Phylogenetic analysis' subsection), within which Kenyan clades were delineated in the previous step (preliminary discrete phylogeographic inference), and whose Kenyan tips were subsampled with the function 'drop.tip' from the R package 'ape' (*Paradis and Schliep, 2019*) according to the above-described subsampling procedure. In order to identify the best-supported lineage transitions events between sampled

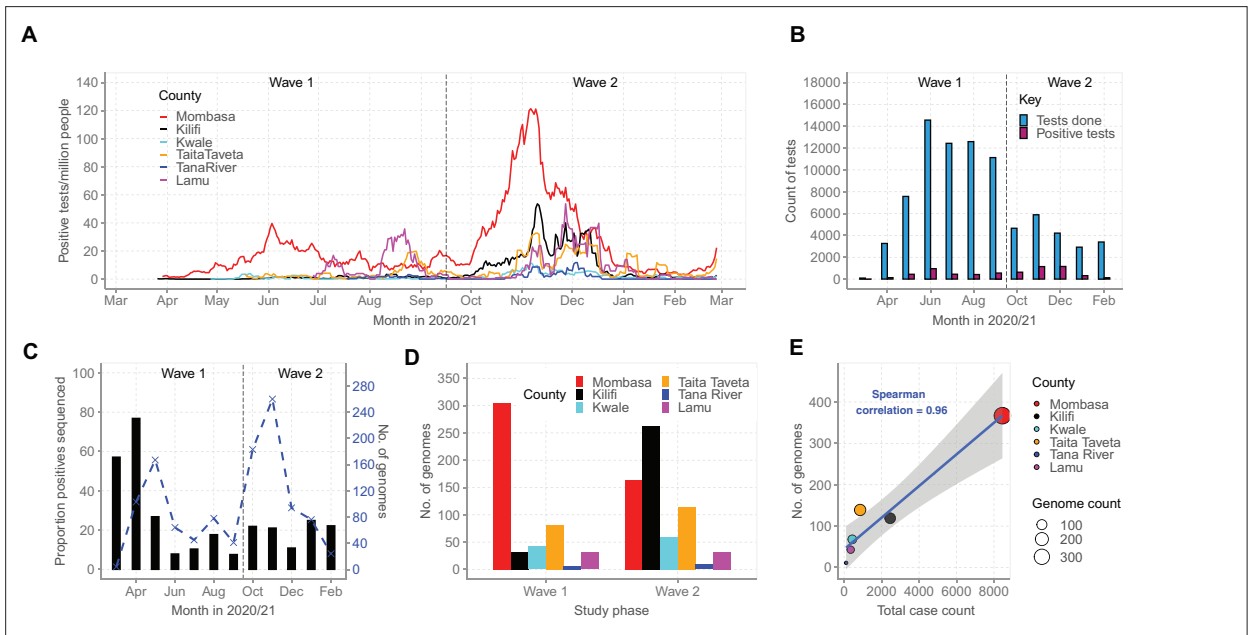

**Figure 2.** Severe acute respiratory syndrome coronavirus 2 (SARS-CoV-2) cases on the Kenyan Coast. (**A**) The epidemic curves for each of the six Coastal Kenya counties derived from the daily positive case numbers, 7-day-rolling average, as reported by the Ministry of Health. (**B**) The monthly count of SARS-CoV-2 RT-PCR tests undertaken at the KEMRI-Wellcome Trust Research Programme (KWTRP) and those positive during the study period. (**C**) The monthly proportion (black bars, primary y-axis) and number (dashed blue line, secondary y-axis) of samples sequenced from total SARS-CoV-2 positives detected at KWTRP. (**D**) County distribution of the sequenced 1139 samples by wave number. (**E**) Linear regression fit of the number of Ministry of Health-reported Coronavirus Disease 2019 (COVID-19) cases in the six Coastal Kenya counties as of February 26, 2021, against the number of SARS-CoV-2 genome sequences obtained at KWTRP during the period.

The online version of this article includes the following source data and figure supplement(s) for figure 2:

**Source data 1.** Number of daily positive tests per million people for each of the six Coastal Kenya counties.

**Source data 2.** Total monthly severe acute respiratory syndrome coronavirus 2 (SARS-CoV-2) tests at KEMRI-Wellcome Trust Research Programme (KWTRP) and identified positives.

**Source data 3.** Monthly proportion of positive samples whole genome sequenced from the positive tests at KEMRI-Wellcome Trust Research Programme (KWTRP).

**Source data 4.** Number of genomes available across the six coastal counties during the two national waves of infections.

**Source data 5.** Total case count and number genomes available from the six coastal counties.

**Figure supplement 1.** Laboratory flow of samples analysed in this study.

coastal counties, we here used the Bayesian stochastic search variable selection (BSSVS) approach (*Lemey et al., 2009*) implemented in BEAST 1.10 (*Suchard et al., 2018*). Each MCMC was run for $10^8$ iterations and sampled every $10^4$ iterations. As described above, MCMC convergence and mixing properties were again inspected with Tracer. Statistical supports associated with transition events connecting each pair of sampled counties were obtained by computing adjusted Bayes factor (BF) supports, that is, BF supports that consider the relative abundance of samples by location (*Dellicour et al., 2021b*; *Vrancken et al., 2021*).

## Epidemiological data

The Kenya daily case data between March 2020 and February 2021 was downloaded from Our World in Data (https://ourworldindata.org/coronavirus/country/kenya; last accessedAugust 4 2021). The daily number of confirmed cases in each county during the study period was obtained from the Kenya Ministry of Health website, which provided the breakdown by county. Metadata for the Coastal Kenya samples was gathered from Ministry of Health case investigation forms delivered together with the samples to KWTRP.

## Kenya COVID-19 response

We derived the overall status of Kenya government COVID-19 interventions using the Oxford Stringency Index (SI) available from Our World in Data database (https://ourworldindata.org/coronavirus/country/kenya, last accessed on January 18, 2022; *Figure 1C*). Oxford SI is based on nine response indicators rescaled to values of 0–100, with 100 being strictest (*Hale et al., 2021*). The nine response indicators used to form the SI are (1) school closures, (2) workplace closures, (3) cancellation of public events, (4) restrictions on public gatherings, (5) closures of public transport, (6) stay-at-home requirements, (7) public information campaigns, (8) restrictions on internal movements, and (9) international travel controls. The various government COVID-19 measures and the dates they took effect or when they were lifted are provided in *Supplementary file 1* and are also reviewed in detail in *Brand et al., 2021*; *Wambua et al., 2022*.

## Statistical analysis

Statistical data analyses were performed in R v4.0.5. Summary statistics (proportions, means, median, and ranges) were inferred where applicable. The 'lm' function in R was used to fit a linear regression model evaluating the relationship between sampling dates and root-to-tip genetic distance in the ML phylogeny. The goodness of fit was inferred from the correlation coefficient. Proportions were compared using chi-square test or Fisher's exact test as appropriate.

## Results

### COVID-19 waves in Coastal Kenya and sequencing at KWTRP

By February 2021, Mombasa, Lamu, and Taita Taveta counties had experienced at least two waves of SARS-CoV-2 infections while Kilifi, Kwale, and Tana River had experienced only a single wave of infections (*Figure 2A*). Up to February 26, 2021, the MoH had reported a cumulative total of 12,655 cases for all the six coastal counties, a majority from Mombasa County (n = 8450, 67%; *Table 1*). Over the same period, KWTRP tested an aggregate of 82,716 NP/OP swabs from the six coastal counties, 6329 (8%) were positive, distributed by month as shown in *Figure 2B*. The majority of the KWTRP positives were from Mombasa County (n = 3139, 50%).

Among the positive cases, we sequenced 1139 cases (18%) distributed by county as reported in *Table 1*. The sample flow is summarized in *Figure 2—figure supplement 1*. The sequenced samples were spread across Wave 1 (n = 499, 44%) and Wave 2 (n = 640, 56%; *Figure 2C and D*) and corresponded to approximately one sequence for every 11 confirmed cases in the region. A high correlation was observed between the MoH case count and the number of samples sequenced for each county ($R^2$ = 0.9216, *Figure 2E*).

### Demographic characteristics of the sequenced sample

The demographic details of the SARS-CoV-2-positive participants identified at KWTRP are presented in *Table 2*. Compared to Wave 1, Wave 2 identified slightly older individuals as positive (median age, 34 vs. 35 years); females were identified as positive more often (26% vs. 32%), Kenyans were identified as positive more often (80% vs. 88%), and fewer individuals with international travel histories

**Table 2.** Demographic characteristics of the positive cases identified at KEMRI-Wellcome Trust Research Programme (KWTRP) in Coastal Kenya by sequencing status and wave period.

| Characteristic | Total positives (n = 6329)(%) | Overall sequencing status | | | Total positives by wave period | | | Total sequenced by wave period | | |
|---|---|---|---|---|---|---|---|---|---|---|
| | | Sequenced (n = 1139)(%) | Non-sequenced (n = 5190)(%) | p-Value[†] | Wave 1 (n = 2849)(%) | Wave 2 (n = 3480)(%) | p-Value[†] | Wave 1 (n = 499)(%) | Wave 2 (n = 640)(%) | p-Value[†] |
| *Age category (years)* | | | | <0.001 | | | 0.0149 | | | 0.0419 |
| 0–9 | 178 (2.8) | 22 (1.9) | 156 (3.0) | | 94 (3.3) | 84 (2.4) | | 11 (2.2) | 11 (1.7) | |
| 10–19 | 472 (7.5) | 85 (7.5) | 387 (7.5) | | 185 (6.5) | 287 (8.2) | | 21 (4.2) | 64 (10.0) | |
| 20–29 | 1682 (26.6) | 234 (20.5) | 1,448 (27.9) | | 769 (27.2) | 913 (26.1) | | 94 (18.9) | 140 (21.8) | |
| 30–39 | 1653 (26.1) | 290 (25.5) | 1,363 (26.3) | | 764 (27.0) | 889 (25.4) | | 123 (24.7) | 167 (26.1) | |
| 40–49 | 1140 (18.0) | 218 (19.1) | 922 (17.8) | | 488 (17.2) | 652 (18.6) | | 88 (17.7) | 130 (20.3) | |
| 50–59 | 605 (9.6) | 122 (10.7) | 483 (9.3) | | 247 (8.7) | 358 (10.2) | | 57 (11.4) | 65 (10.1) | |
| 60–69 | 187 (2.9) | 46 (4.0) | 141 (2.7) | | 78 (2.8) | 109 (3.1) | | 23 (4.6) | 23 (3.6) | |
| 70–79 | 74 (1.1) | 17 (1.5) | 57 (1.1) | | 33 (1.2) | 41 (1.2) | | 7 (1.4) | 10 (1.6) | |
| 80+ | 13 (0.2) | 4 (0.4) | 9 (0.2) | | 7 (0.2) | 6 (0.2) | | 3 (0.6) | 1 (0.2) | |
| Missing | 325 (3.25) | 101 (8.9) | 224 (4.3) | | 167 (5.9) | 158 (4.5) | | 71 (14.3) | 30 (4.7) | |
| *Gender* | | | | 0.554 | | | <0.001 | | | 0.1979 |
| Female | 1896 (29.9) | 333 (29.2) | 1563 (30.1) | | 763 (26.9) | 1,133 (32.4) | | 125 (25.1) | 208 (32.4) | |
| Male | 4058 (64.1) | 686 (60.2) | 3372 (65.0) | | 1860 (65.7) | 2198 (62.9) | | 288 (57.8) | 398 (62.1) | |
| Missing | 375 (5.9) | 120 (10.5) | 255 (4.9) | | 209 (7.4) | 166 (4.7) | | 85 (17.1) | 85 (5.5) | |
| *Nationality* | | | | <0.001 | | | <0.001 | | | <0.001 |
| Kenyan | 5356 (84.6) | 870 (76.4) | 4486 (86.4) | | 2270 (80.2) | 3086 (88.2) | | 316 (63.5) | 554 (86.4) | |
| Tanzania | 131 (2.1) | 34 (3.0) | 97 (1.9) | | 81 (2.9) | 50 (1.4) | | 25 (5.0) | 9 (1.4) | |
| Uganda | 16 (0.3) | 1 (0.1) | 15 (0.3) | | 10 (0.4) | 6 (0.2) | | 0 (0.2) | 4 (0.0) | |
| *Ethiopia* | 14 (0.2) | 4 (0.4) | 10 (0.2) | | 0 (0.0) | 14 (0.4) | | 1 (0.2) | 0 (0.0) | |
| Other† | 117 (1.84) | 24 (2.1) | 93 (1.8) | | 46 (1.6) | 71 (2.0) | | 6 (1.2) | 18 (2.8) | |
| Missing | 695 (10.9) | 206 (18.1) | 489 (9.4) | | 425 (15.0) | 270 (7.7) | | 150 (30.1) | 56 (8.7) | |
| *Travel history** | | | | <0.001 | | | <0.001 | | | <0.001 |
| Yes | 485 (7.7) | 119 (10.4) | 366 (7.1) | | 340 (12.0) | 145 (4.1) | | 83 (16.7) | 36 (5.6) | |
| No | 2562 (40.7) | 407 (35.7) | 2155 (41.5) | | 1372 (48.4) | 1190 (34.0) | | 189 (38.0) | 218 (34.0) | |
| Missing | 3282 (51.9) | 613 (53.8) | 2669 (51.4) | | 1120 (39.5) | 2162 (61.8) | | 226 (45.4) | 387 (60.4) | |

*Defined as having moved into Kenya in the previous 14 days or sampled at a point of entry (POE) into Kenya.

[†]p-value calculated using a Pearson's chi-squared test, for variables where some cells in the table had <5 observations, Fishers' exact test was applied.

were identified as positive (12% vs. 4%). Tanzania ranked second in terms of the number of individuals providing sequenced samples (n = 34, 4%). A total of 119 samples (15%) were sequenced from people who had recently travelled internationally (within 14 days). Travel history information was missing for 613 (54%) sequenced cases (*Table 2*).

## Viral lineages circulating in Coastal Kenya

The 1139 Coastal Kenya genomes were classified into 43 Pango lineages, including 4 first identified in Kenya (N.8, B.1.530, B.1.549, and B.1.596.1) and 2 global variants of concern (VOC); B.1.1.7 (Alpha) and B.1.351 (Beta; *Table 3*). A total of 23 and 29 lineages were observed during Wave 1 and Wave 2, respectively, with 9 lineages detected in both waves (*Figure 3A and B*). Nineteen lineages were identified in three or more samples with the top six lineages accounting for 89% of the sequenced infections, namely, B.1 (n = 723, 63%), B.1.549 (n = 143, 13%), B.1.1 (n = 57, 5%), B.1.530 (n = 32,

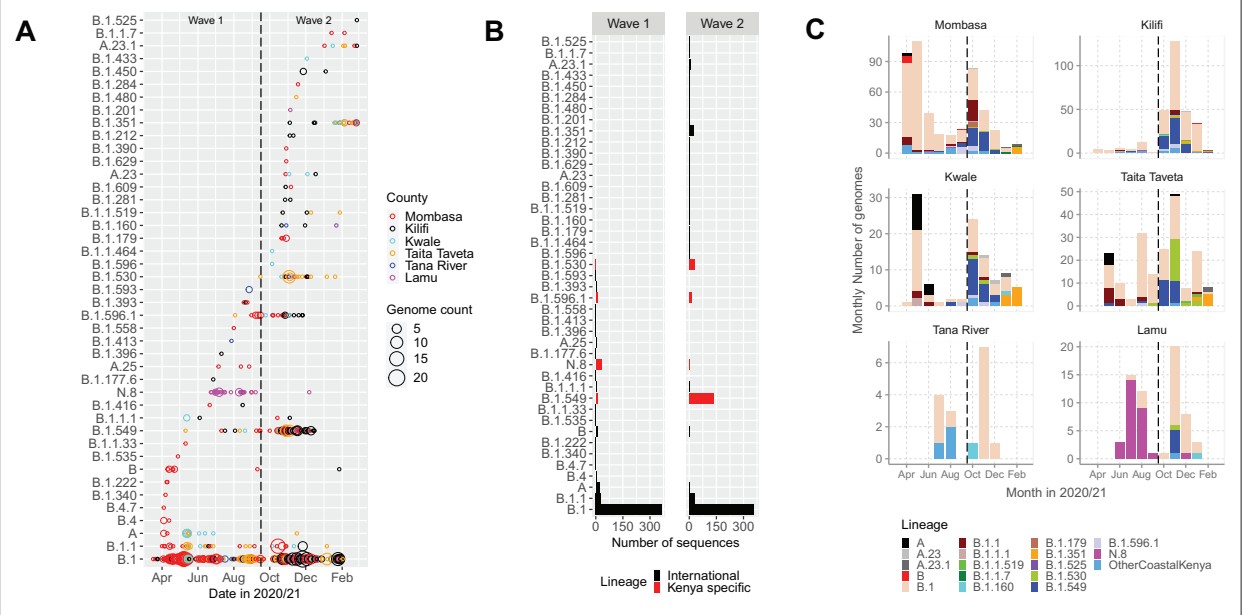

**Figure 3.** Lineage introductions and temporal dynamics in Coastal Kenya. (**A**) Timing of detections of severe acute respiratory syndrome coronavirus 2 (SARS-CoV-2) Pango lineages in the sequenced 1139 Coastal Kenya samples. The circle size scaled by number of daily detections. The vertical dashed line demarcates the date of transition from Wave 1 to Wave 2. (**B**) Cumulative detections by Pango lineage detections by wave number. The bars are coloured by known information about the lineages; Kenya specific (B.1.530, B.1.549, B.1.596.1, and N.8, red bars) or international lineages (black bars). (**C**) Monthly distribution of the common lineages identified across the six counties presented as raw counts of the sequenced infections. Lineages detected in less than four cases or not considered a variant of concern (VOC) or variant of interest (VOI) were put together and referred to as 'other Coastal Kenya lineages'. This group comprises 26 lineages, namely, A.25, B.1.1.33, B.1.1.464, B.1.177.6, B.1.201, B.1.212, B.1.222, B.1.281, B.1.284, B.1.340, B.1.390, B.1.393, B.1.396, B.1.413, B.1.416, B.1.433, B.1.450, B.1.480, B.1.535, B.1.558, B.1.593, B.1.596, B.1.609, B.1.629, B.4, and B.4.7.

The online version of this article includes the following source data for figure 3:

**Source data 1.** The total daily number of sequenced cases for each identified lineage across each of the six coastal counties.

**Source data 2.** Total cases sequenced for each 43 identified lineages in the two waves of infection in Kenya.

**Source data 3.** The monthly number of cases for each lineage across the two waves of infection in Kenya.

3%), N.8 (n = 31, 3%), and B.1.351 (n = 26, 2%; *Table 3*). Many of the lineages were first detected in Mombasa (n = 21, 49%) before observation in other counties (*Supplementary file 3*). The temporal pattern of detection for the lineages across six counties is shown in *Figure 3C*.

We detected an average of eight Pango lineages in circulation per month during the study period; the lowest (n = 1) in March 2020 and the highest (n = 17) in November 2020 (*Figure 4*). The earliest sequences for 7 lineages (16%) came from individuals who reported recent international travel while earliest sequences for 16 lineages (37%) came from individuals who had no history of recent travel, and the earliest sequences for 20 lineages (47%) came from individuals who had no information about travel history (*Figure 4—figure supplement 1*). Among the individuals with recent travel history, the top five lineages were B.1, A, B.1.1, B.1.549, and B.1.351 (*Figure 4—figure supplement 2*). Most of the lineages detected in Coastal Kenya were first detected in Mombasa County (n = 14, 58%; *Supplementary file 3*).

## SARS-CoV-2 lineage dynamics beyond Coastal Kenya

We evaluated various scales of observation to illustrate the spatial-temporal lineage dynamics during our study period (*Figure 5*). The genome set was carefully selected to minimize sampling bias (*Figure 5—figure supplement 1*). A total of 33 Pango lineages were identified for the Kenya sample, 125 lineages for Eastern Africa, 337 lineages for Africa, and 950 lineages globally (*Supplementary file 4*). The number of lineages detected for the different scales was consistent with the widening scope except for across Kenya where a relatively small number of genomes were available. The top 10 Pango lineages observed at each scale of observation is provided in *Supplementary file 5*.

**Table 3.** Lineages observed in Coastal Kenya, their county distribution, global history, and variants of concern (VOC)/variants of interest (VOI) status.

| Lineage | Frequency (%) | Mombasa | Kilifi | Kwale | Taita Taveta | Tana River | Lamu | Earliest date | Number assigned | Description |
|---|---|---|---|---|---|---|---|---|---|---|
| | | | | | | | | | | Root of the pandemic lies within lineage A |
| A | 22 (0.3) | 3 | - | 13 | 6 | - | - | Decmber 30, 2019 | 2224 | Predominantly found in China |
| A.23 | 4 (0.1) | 1 | 1 | 2 | - | - | - | August 14, 2020 | 92 | Predominantly found in Uganda |
| A.23.1 | 6 (0.1) | 2 | 1 | 1 | 2 | - | - | September 21, 2020 | 1191 | International lineage |
| A.25 | 3 (0.0) | 3 | - | - | - | - | - | June 8, 2020 | 47 | Predominantly found in Uganda |
| B | 9 (0.1) | 8 | 1 | - | - | - | - | December 24, 2019 | 7358 | Second major haplotype (and first to be discovered) |
| B.1 | 723 (11.4) | 328 | 192 | 44 | 119 | 12 | 28 | January 1, 2020 | 88,731 | Predominantly found in Europe, origin corresponds to the Northern Italian outbreak early in 2020 |
| B.1.1 | 57 (0.9) | 33 | 6 | 5 | 13 | - | - | January 8, 2020 | 49,562 | Predominantly found in Europe |
| B.1.1.1 | 5 (0.1) | 1 | 2 | 2 | - | - | - | March 2, 2020 | 2827 | Predominantly found in England |
| B.1.1.33 | 1 (0.0) | 1 | - | - | - | - | - | March 1, 2020 | 2117 | Predominantly found in Brazil |
| B.1.1.464 | 1 (0.0) | - | - | 1 | - | - | - | April 1, 2020 | 666 | Predominantly found in USA |
| B.1.1.519 | 4 (0.1) | - | 2 | - | 2 | - | - | July 30, 2020 | 23,815 | Predominantly found in USA/ Mexico |
| B.1.1.7 | 2 (0.0) | 2 | - | - | - | - | - | September 3, 2020 | 1,062,326 | Alpha variant of concern |
| B.1.160 | 5 (0.1) | - | 2 | 1 | - | 1 | 1 | February 2, 2020 | 28,128 | Predominantly found in Europe |
| B.1.177.6 | 1 (0.0) | - | 1 | - | - | - | - | May 29, 2020 | 949 | Predominantly found in Wales |
| B.1.179 | 5 (0.1) | 5 | - | - | - | - | - | March 9, 2020 | 242 | Predominantly found in Denmark |
| B.1.201 | 1 (0.0) | - | - | - | - | - | 1 | March 6, 2020 | 173 | Predominantly found in the UK |
| B.1.212 | 2 (0.0) | - | 2 | - | - | - | - | March 3, 2020 | 59 | Predominantly found in South America |
| B.1.222 | 2 (0.0) | 2 | - | - | - | - | - | February 24, 2020 | 568 | Predominantly found in Scotland |
| B.1.281 | 2 (0.0) | - | 2 | - | - | - | - | April 8, 2020 | 41 | Predominantly found in Bahrain |
| B.1.284 | 1 (0.0) | 1 | - | - | - | - | - | March 9, 2020 | 85 | Predominantly found in TX,USA |
| B.1.340 | 1 (0.0) | 1 | - | - | - | - | - | March 13, 2020 | 221 | Predominantly found in USA |
| B.1.351 | 26 (0.4) | 6 | 5 | 8 | 7 | - | - | September 1, 2020 | 29,720 | Beta variant of concern |
| B.1.390 | 1 (0.0) | 1 | - | - | - | - | - | March 25, 2020 | 91 | Predominantly found in USA |
| B.1.393 | 3 (0.0) | 2 | 1 | - | - | - | - | May 29, 2020 | 34 | Predominantly found in Uganda |
| B.1.396 | 1 (0.0) | - | 1 | - | - | - | - | April 6, 2020 | 1375 | Predominantly found in USA |
| B.1.413 | 1 (0.0) | - | - | - | - | 1 | - | March 12, 2020 | 195 | Predominantly found in USA |
| B.1.416 | 2 (0.0) | 1 | 1 | - | - | - | - | April 11, 2020 | 594 | Predominantly found in Senegal/ Gambia, reassigned from B.1.5.12 |
| B.1.433 | 1 (0.0) | - | - | 1 | - | - | - | August 3, 2020 | 314 | Predominantly found in TX, USA |
| B.1.450 | 3 (0.0) | - | 3 | - | - | - | - | March 14, 2020 | 86 | Predominantly found in TX, USA |
| B.1.480 | 1 (0.0) | - | - | - | 1 | - | - | July 3, 2020 | 386 | Predominantly found in England, Australia, Sweden, Norway |
| B.1.525 | 1 (0.0) | - | 1 | - | - | - | - | March 28, 2020 | 8012 | Eta variant of interest |
| B.1.530 | 32 (0.5) | 3 | 4 | 2 | 22 | - | 1 | October 1, 2020 | 111 | Predominantly found in Kenya |
| B.1.535 | 1 (0.0) | 1 | - | - | - | - | - | March 22, 2020 | 29 | Predominantly found in Australia |
| B.1.549 | 143 (2.3) | 42 | 56 | 18 | 23 | - | 4 | May 11, 2020 | 171 | Predominantly found in Kenya and England |

*Table 3 continued on next page*

*Table 3 continued*

| Lineage | Frequency (%) | Mombasa | Kilifi | Kwale | Taita Taveta | Tana River | Lamu | Earliest date | Number assigned | Description |
|---|---|---|---|---|---|---|---|---|---|---|
| B.1.558 | 1 (0.0) | 1 | - | - | - | - | - | April 6, 2020 | 211 | Predominantly found in USA/ Mexico |
| B.1.593 | 2 (0.0) | - | - | - | - | 2 | - | July 3, 2020 | 99 | Predominantly found in USA |
| B.1.596 | 1 (0.0) | - | - | 1 | - | - | - | April 11, 2020 | 9968 | Predominantly found in USA |
| B.1.596.1 | 24 (0.4) | 12 | 8 | 3 | 1 | - | - | September 7, 2020 | 83 | Predominantly found in Kenya |
| B.1.609 | 2 (0.0) | 1 | 1 | - | - | - | - | March 10, 2020 | 1879 | Predominantly found in USA/ Mexico |
| B.1.629 | 1 (0.0) | 1 | - | - | - | - | - | July 12, 2020 | 231 | Lineage circulating in several countries |
| B.4 | 3 (0.0) | 3 | - | - | - | - | - | January 18, 2020 | 386 | Predominantly found in Iran |
| B.4.7 | 1 (0.0) | 1 | - | - | - | - | - | March 14, 2020 | 68 | Predominantly found in Africa and UAE |
| N.8 | 31 (0.5) | 2 | 1 | - | - | - | 28 | June 23, 2020 | 15 | Alias of B.1.1.33.8, predominantly found in Kenya |

By January 2021, the lineages B.1.1.7 and B.1.351 were already widely spread across Eastern Africa and Africa but there were only sporadic detections in Coastal Kenya (*Figure 5A–D*). Waves 1 and 2 Coastal Kenya predominant lineage B.1 occurred in substantial proportions across the different scales early in the pandemic (Wave 1), but its prevalence elsewhere outside Kenya diminished faster overtime compared to the Kenya sample. Greater than 95% (909/950) of the lineages comprising infections in the global subsample (March 1, 2020, and February 28, 2021) were not seen in the Coastal Kenya samples (*Supplementary file 5*). The global pattern of detection of the 43 locally detected lineages is shown in *Figure 5—figure supplement 2*. Only two lineages in the Coastal Kenya sampling were not in the global subsample; lineage N.8 and lineage B.1.593 (*Figure 5—figure supplement 2*).

## SARS-CoV-2 genetic diversity in Coastal Kenya

A time-resolved ML phylogeny for the Coastal Kenya genomes with global subsample in the background is provided in *Figure 6*. This phylogeny showed that (1) the Coastal Kenya genomes were

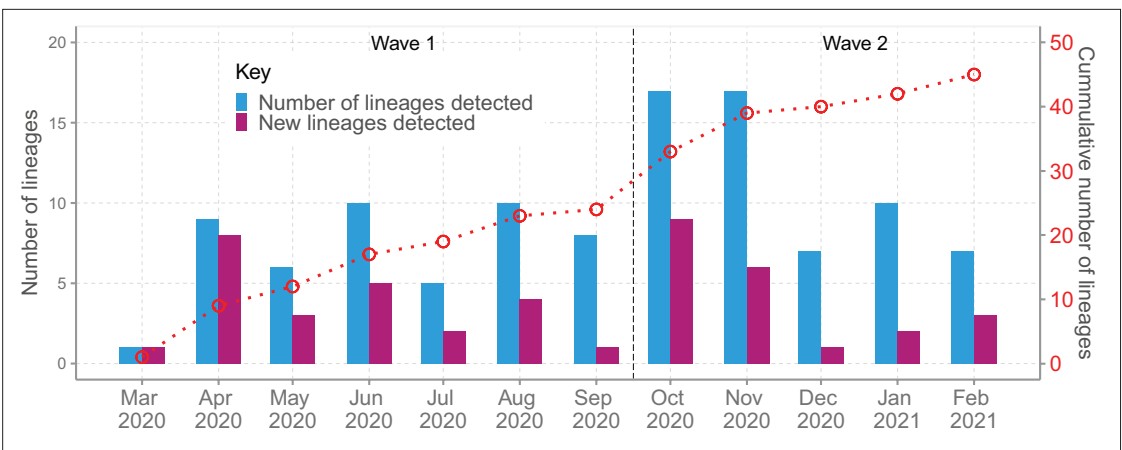

**Figure 4.** Lineage detection patterns in Coastal Kenya showing monthly count of total detected lineages, detected new lineages, and commutative total of detected lineages in Coastal Kenya across the study period (secondary axis).

The online version of this article includes the following source data and figure supplement(s) for figure 4:

**Source data 1.** New, total circulating and cumulative Pango lineage counts by month in Coastal Kenya.

**Source data 2.** Distribution of the detected Pango lineages by travel history information in Coastal Kenya.

**Figure supplement 1.** Demographic characteristics of the sequenced index cases of the 43 lineages identified in Coastal Kenya.

**Figure supplement 2.** Lineages detected among individuals who reported a recent international travel history (n = 119) and their distribution by nationality.

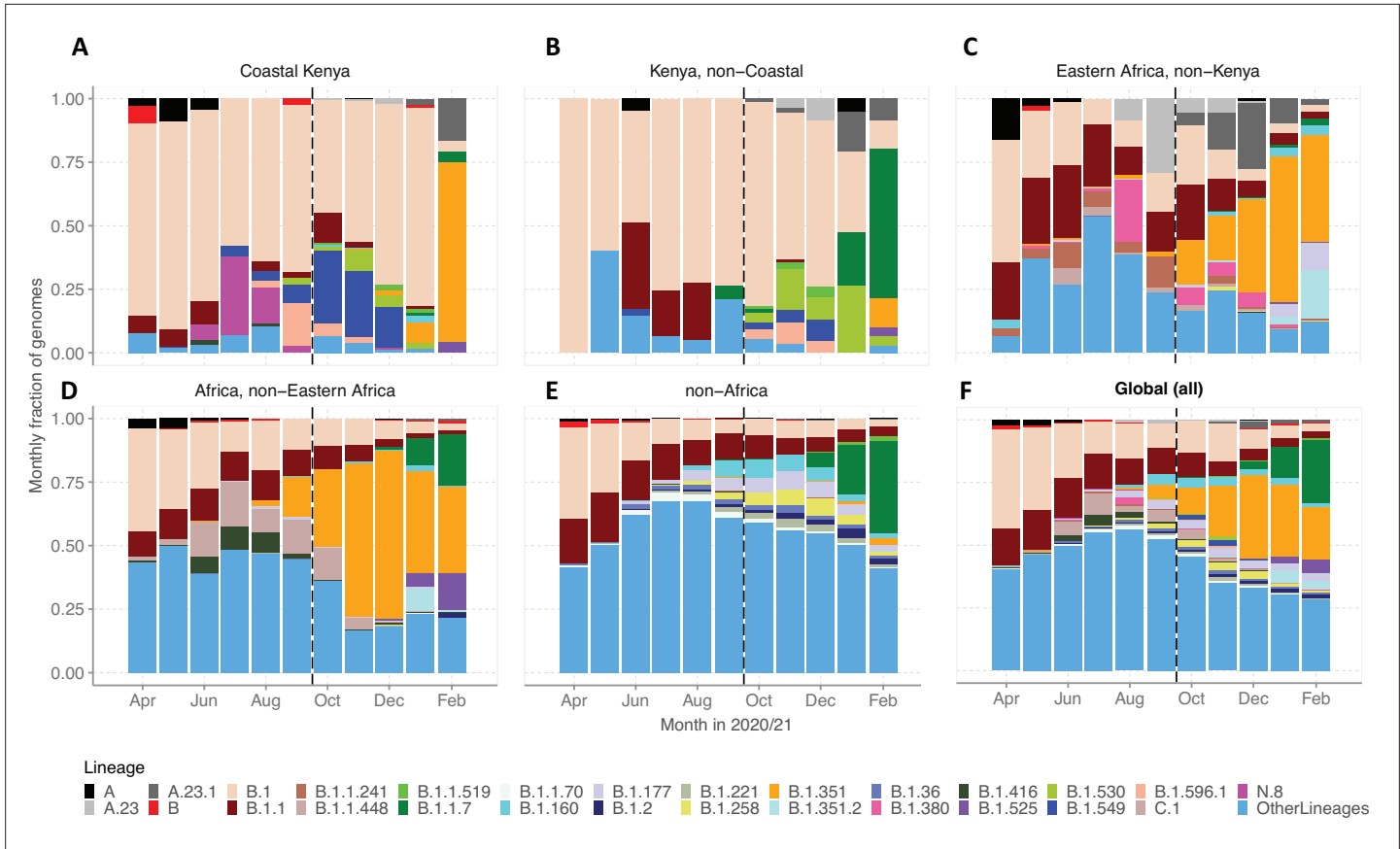

**Figure 5.** Investigation of lineage spatial temporal dynamics at widening scales of observation. (**A**) Monthly prevalence of detected lineages in Coastal Kenya from the sequenced 1139 genomes. (**B**) Monthly prevalence of detected lineages in Kenya (outside coastal counties) from 605 contemporaneous genomes data is available in GISAID. (**C**) Monthly prevalence of detected lineages in Eastern Africa from 3531 contemporaneous genomes from 10 countries whose contemporaneous data are available in GISAID. The included countries were Comoros, Ethiopia, Madagascar, Malawi, Mozambique, Reunion, Rwanda, Uganda, Zambia, and Zimbabwe. (**D**), Monthly distribution of detected lineages in African countries (excluding Eastern Africa). A total of 14,874 contemporaneous genomes from 37 countries that were available in GISAID are included in the analysis. (**E**) Monthly prevalence of detected lineages in a global subsample of 19,993 contemporaneous genomes from 147 countries that were compiled from GISAID (see detail in 'Methods' section). Genomes from African samples are excluded in this panel. (**F**) Includes all genomes analysed from the scales (**A–E**). Lineages not among the top 10 in at least one of the five scales of observation investigated have been lumped together as 'Other lineages'.

The online version of this article includes the following source data and figure supplement(s) for figure 5:

**Source data 1.** Monthly counts for the top lineages observed at the different scales of observation analysed.

**Figure supplement 1.** Flow of the genomes retrieved from GISAID and used in the comparative genomic epidemiology for lineage dynamics analysis and global context of the Coastal Kenya genomes.

**Figure supplement 2.** Global context and temporal dynamics of the Pango lineages detected in Coastal Kenya.

represented across several but not all of the major phylogenetic clusters, (2) some of the Coastal Kenya clusters mapped into known Pango lineages, some of which appeared to expand after introduction, and (3) all six coastal counties appeared to have each had multiple virus introductions with some of the clusters comprising genomes detected across multiple counties (*Figure 6*). Many of the lineages identified in Coastal Kenya formed monophyletic groups (e.g. A, B.1.549, B.1.530, and N.8) with a few exceptions like lineage B.1, B.1.1, and B.1.351 which occurred on the phylogeny as multiple clusters. The data we analysed showed considerable correlation between the root-to-tip genetic distance and the sampling dates of the genomes ($R^2$ = 0.604; *Figure 6—figure supplement 1*).

We found that sequences from individuals reporting recent travel (n = 119) occurred throughout the local phylogeny based on the clustering of the Coastal Kenya genomes (*Figure 6—figure supplement 2*). Recent travellers infected with lineage B.1 (n = 60, 8%) were spread throughout the phylogeny and were captured in all the six counties of Coastal Kenyan counties. Contrastingly,

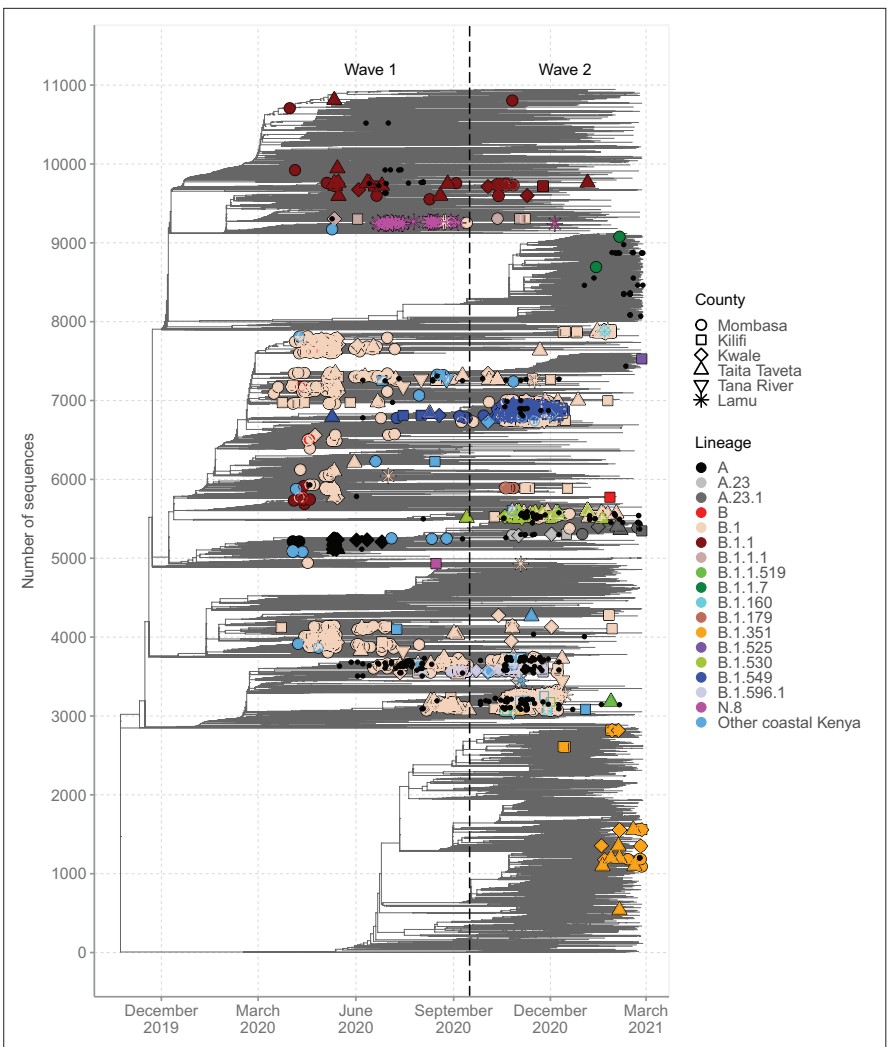

**Figure 6.** Global context of severe acute respiratory syndrome coronavirus 2 (SARS-CoV-2) diversity observed in Coastal Kenya. A time-resolved global phylogeny that combined 1139 Coastal Kenya SARS-CoV-2 genomes and 9906 global reference sequences. Distinct shapes are used to identify the different Coastal Kenya counties and distinct colours to identify the different lineages. Lineages detected in less than four cases were put together and referred to as 'other Coastal Kenya lineages'. This group comprises 26 lineages, namely, A.25, B.1.1.33, B.1.1.464, B.1.177.6, B.1.201, B.1.212, B.1.222, B.1.281, B.1.284, B.1.340, B.1.390, B.1.393, B.1.396, B.1.413, B.1.416, B.1.433, B.1.450, B.1.480, B.1.535, B.1.558, B.1.593, B.1.596, B.1.609, B.1.629, B.4, and B.4.7. Sequences not fitting clock-like molecular evolution were removed using TreeTime program (**Sagulenko et al., 2018**). The analysis included 292 genomes obtained from samples collected in Kenya but outside coastal counties and these are shown as a small, solid black circles.

The online version of this article includes the following figure supplement(s) for figure 6:

**Figure supplement 1.** Root-to-tip regression analysis of Coastal Kenya genomes combined with global sequences.

**Figure supplement 2.** Mutation-resolved maximum likelihood (ML) phylogeny of the 1139 Coastal Kenya genomes annotated with available epidemiological information.

---

individuals reporting recent travel and infected with lineage A (n = 19, 86%) and some of the lineage B.1.1 (n = 10, 18%)-infected cases clustered, suggesting a potential common infection source/origin for these lineages. Viral sequences from Kenyan nationals were spread across the tree structure. One striking exception was lineage A-infected cases whose nationality was frequently recorded as missing, but majority were travellers.

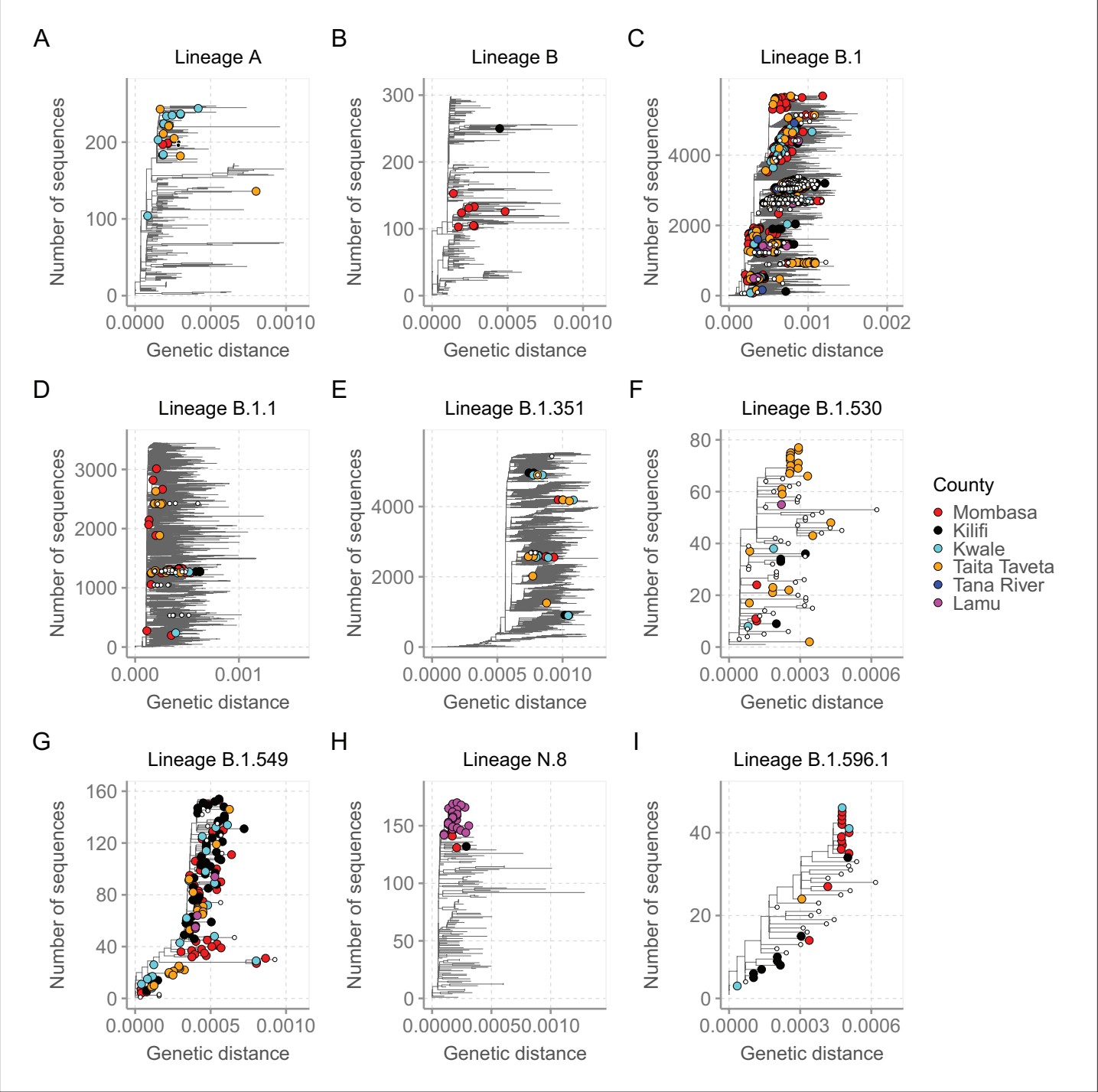

**Figure 7.** Mutation-resolved lineage-specific phylogenies for the top nine lineages detected in Coastal Kenya. The Coastal Kenya genomes are indicated with filled different shapes for the different counties. Genomes from other locations within Kenya are indicated with small solid black circles. (**A**) Phylogeny of the 22 lineage A Coastal Kenya genome combined 240 global lineage A sequences. (**B**) Phylogeny of the lineage B that combined 9 Coastal Kenya genomes and 291 global lineage B sequences. (**C**) Phylogeny for lineage B.1 that combined 723 Coastal Kenya genomes and 5136 global lineage B.1 sequences. (**D**) Phylogeny for lineage B.1.1 that combined 57 Coastal Kenya genomes and 3451 global lineage B.1.1sequences. (**E**) Phylogeny for lineage B.1.351 that combined 26 Coastal Kenya genomes and 5613 global lineage B.1.351 sequences. (**F**) Phylogeny for lineage B.1.530 that combined 32 Coastal Kenya genomes and 45 global lineage B.1.530 sequences. (**G**) Phylogeny for lineage B.1.549 that combined 143 Coastal Kenya genomes and 14 lineage B.1. 549 sequences from other locations. (**H**) Phylogeny for lineage N.8 that combined 31 Coastal Kenya genomes of lineage N.8, a single Coastal Kenya genomes of lineage B.1.1.33 and 139 lineage B.1.1.33 global sequences. (**I**) Phylogeny for lineage B.1.596.1 that combined 24 Coastal Kenya genomes and 22 lineage B.1.596.1 global sequences.

*Figure 7 continued on next page*

*Figure 7 continued*

The online version of this article includes the following figure supplement(s) for figure 7:

**Figure supplement 1.** Time-resolved lineage-specific phylogenetic trees for top nine lineages detected in Coastal Kenya.

**Figure supplement 2.** Genome maps of the 'Kenya' lineages where the spike region is shown in detail and in colour and the rest of the genome is shown in grey colour.

For detailed investigation into the local SARS-CoV-2 genetic diversity, we reconstructed mutation-resolved phylogenies for the top nine lineages in Coastal Kenya (*Figure 7*, and corresponding time-resolved phylogenies presented in *Figure 7—figure supplement 1*). We observed (1) considerable within-lineage diversity (highest in the predominant lineage B.1), (2) formation of multiple subclusters within these lineages, with some of clusters being county-specific (e.g. cluster of Taita Taveta sequences observed in lineage B.1.530; *Figure 7F*), and (3) scenarios of local sequences interspersed with global comparison genomes from the same lineage implying multiple import events of these lineages into Kenya, for example, for lineages A, B, B.1, B.1.1, and B.1.351 (*Figure 7A–E*). Of the four lineages that appeared to be Kenya specific, three (B.1.530, B.1.549, and B.1.596.1) had representation in other parts of Kenya outside of the coastal counties with formation of multiple genetic subclusters (*Figure 7F and I*). However, lineage N.8, which was mainly detected in Lamu, formed a single monophyletic group (*Figure 7H*) when co-analysed with its precursor lineage B.1.1.33.

## Imports and exports from Coastal Kenya

We used ancestral location state reconstruction of the dated phylogeny (*Figure 6*) to infer virus import and export (*Sagulenko et al., 2018*). By this approach, a total of 280 and 105 virus importation and virus exportation events were detected, respectively (*Table 4*), and distributed between the waves as summarized in *Figure 8A and B*. Virus importations and exportations into the region occurred predominantly through Mombasa (n = 140, 50%) and (n = 85, 81%), respectively. However, relative to its population size, Mombasa was second to Taita Taveta in importation rate per 100,000 people (*Table 4*). The majority of the international importation events we detected occurred during Wave 1 (*Figure 8B*). For the detected 105 virus exportations, 71 (68%) occurred during Wave 1 and 34 (32%) during Wave 2 (*Figure 8A and B*). We repeated the analysis using the second global subsample with a normalized subsample of the Coastal Kenya genomes accounting for total reported infections per county. The reanalysis found closely aligned results to those revealed by subsample1 (*Supplementary file 6*).

## Viral circulation between counties of Coastal Kenya

To explore the pattern of viral circulation within and among counties of Coastal Kenya, we conducted replicated discrete phylogeographic analyses based on random subsets of genomic sequences subsampled according to local incidence (*Figure 9*). We observe notable differences among the reconstructions of viral lineage dispersal history obtained from the 10 replicated analyses, meaning that the phylogeographic outcome is quite sensitive to the sampling pattern. However, if we look at the

**Table 4.** Summary of import and export events and rates into coastal counties populations.

| County | Virus import (%) | Import rate (per 100,000)* | Virus export (%) | Export rate (per 100,000)* |
|---|---|---|---|---|
| Mombasa | 140 (50) | 11.6 | 85 (81) | 7.0 |
| Kilifi | 53 (19) | 3.6 | 4 (4) | 0.3 |
| Kwale | 33 (12) | 3.8 | 4 (4) | 0.5 |
| Taita Taveta | 46 (16) | 13.5 | 12 (11) | 3.5 |
| Tana River | 2 (<1) | 0.6 | - | - |
| Lamu | 6 (2) | 4.1 | - | - |
| Overall | 280 | 6.7 | 105 | 2.4 |

*Denominator population as per the 2019 national census (see *Table 1*).

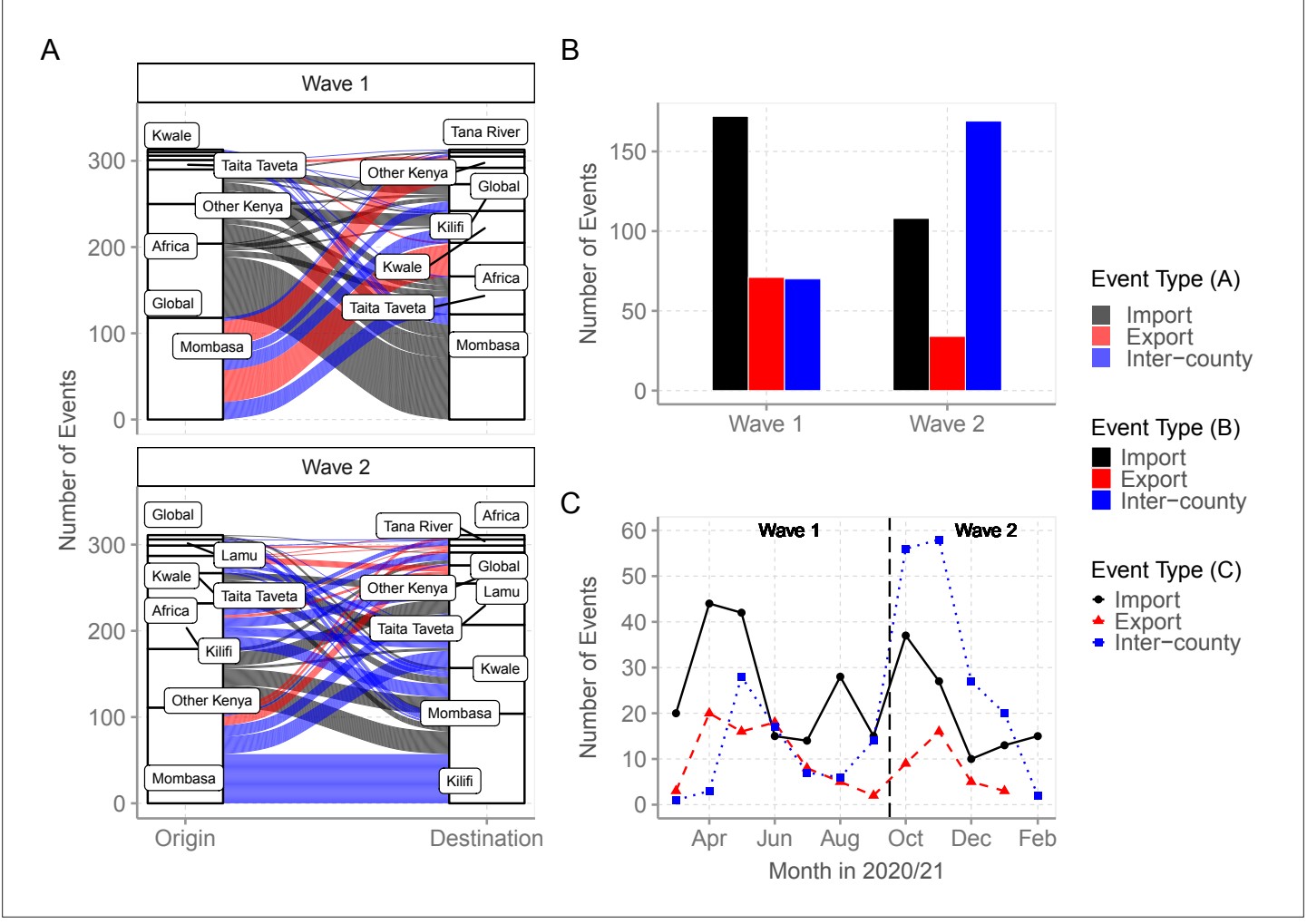

**Figure 8.** Virus importations and exportations from Coastal Kenya. (**A**) Alluvium plots stratified by wave number showing the estimated number and flow of importations into and exportations from Coastal Kenya. 'Global' refer to origins or destinations outside Kenya while 'Other Kenya' refer to origins or destinations within Kenya but outside the Coastal Counties. (**B**) The raw counts bar plot of location transition events observed within and between Coastal Kenya outside world shown as either virus exportations, importations, or inter-county transmission, these stratified by wave number. (**C**) Monthly trends of the observed transition events stratified by type. The findings presented in this figure are based on subsample 1.

The online version of this article includes the following source data and figure supplement(s) for figure 8:

**Source data 1.** The number of importation and exportation events by county and wave period.

**Source data 2.** The number of importations, inter-county transmission, and exportation events by month.

**Figure supplement 1.** The number of import events, stratified by wave period into the individual coastal counties and their estimated origins.

similarities among those replicated phylogeographic reconstruction, we can observe that Mombasa tended to act as an important hub associated with relatively important viral circulation and at the origin of numbers of viral dispersal events toward surrounding counties.

## Discussion

We report patterns of SARS-CoV-2 introduction and spread in Coastal Kenya during Waves 1 and 2, and estimate approximately 300 independent virus introductions occurred, many in the first six months of the pandemic. Given the limited diagnostic testing capacity and the relatively small number of samples sequenced, it is likely that there were more introductions than calculated here.

Multiple virus introductions occurred even at the county level, with inter-county spread predominating Wave 2. A lockdown was put in place for Mombasa, Kilifi, and Kwale in April 2020 and was later

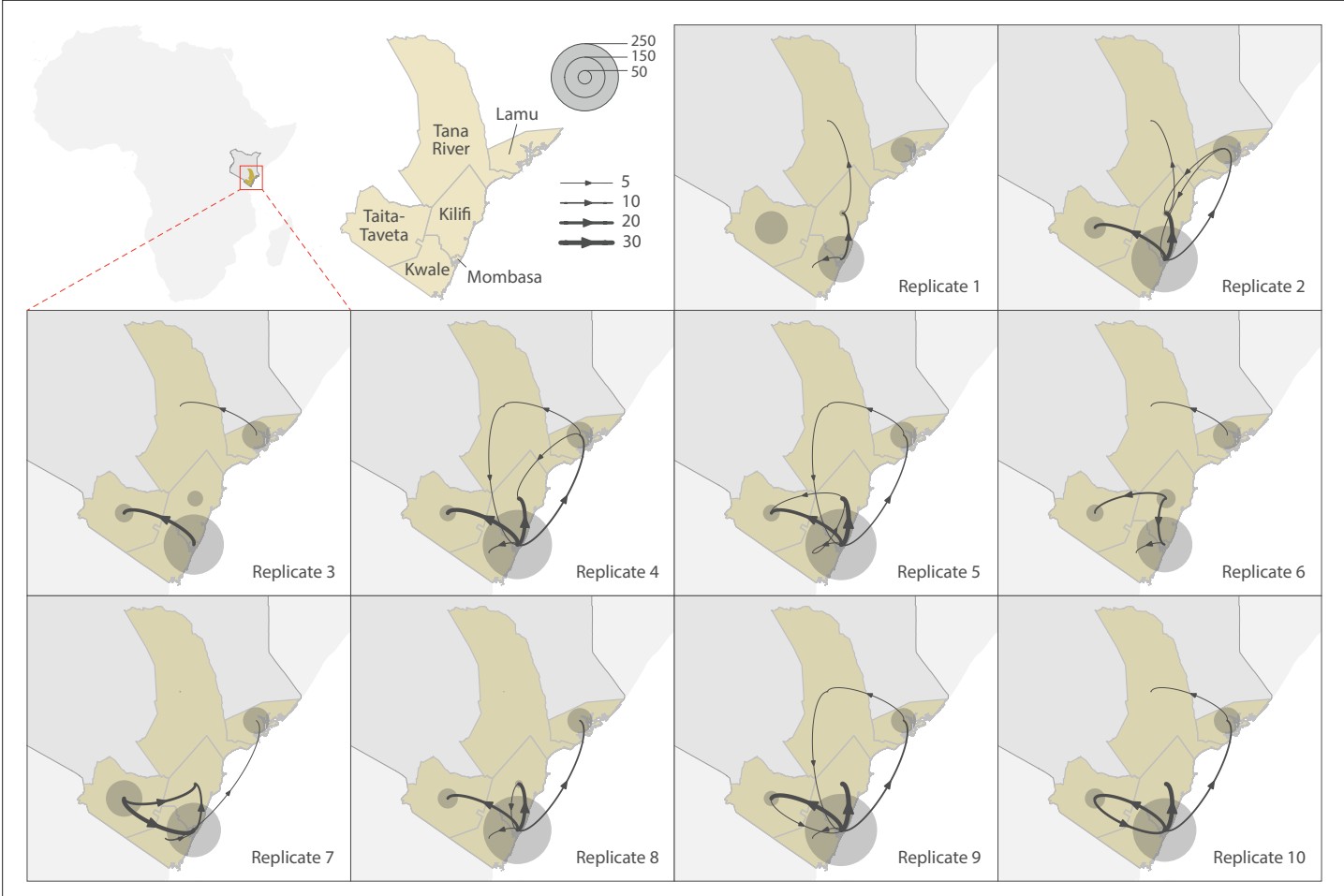

**Figure 9.** Replicated discrete phylogeographic reconstructions of the circulation of severe acute respiratory syndrome coronavirus 2 (SARS-CoV-2) lineages within and among counties of Coastal Kenya. Each replicated analysis was based on a random subset of genomic sequences subsampled according to local incidence (see the 'Methods' section for further detail). We here report the number of lineage dispersal events inferred among (arrows) and within (transparent grey circles) counties, both measures being averaged over posterior trees sampled from each posterior distribution. We here only report among-counties transition events supported by adjusted Bayes factor (BF) values >20, which corresponds to a strong support according to the scale of BF values interpretation of **Kass and Raftery, 1995**.

lifted on June 7, 2020, allowing mixing of the population and potential virus spread. It is notable that most imports into and exports from the Coastal Region probably passed through Mombasa, a major commercial, industrial, and tourist destination. This observation highlights the need for continuous and systematic surveillance of lineages circulating in Mombasa timely knowledge of variants entering or circulating within Coastal Kenya.

During Wave 1, we detected 23 Pango lineages in Coastal Kenya with lineage B.1 accounting for 73% of the sequenced infections. B.1 was detected in all counties of Coastal Kenya and was considerably diverse. Lineage B.1 dominance may have been in part driven by the possession of the D614G change in the spike protein, which has been found to enhance viral fitness (**Baric, 2020**). The strict quarantine and isolation of confirmed cases in the early period may have prevented some of the other lineages introduced from widely spreading, for example, lineage A was limited to travellers.

Lineage N.8 was specific to Lamu County with only three cases recorded elsewhere in Coastal Kenya and three cases elsewhere in Kenya. Lineage N.8 precursor (lineage B.1.1.33) was observed earlier in Brazil. The occurrence of lineage N.8 in Lamu may have arisen from its direct introduction from outside Kenya or introduction as B.1.1.33 followed by local evolution. Determining the exact origin of this lineage is complicated by the sparse genomic surveillance elsewhere Kenya during the study period and indeed for many regions across the world. The N.8 lineage has seven characteristic

lineage defining mutations including S: D614G and N: R203K, N: G204R, and N: I292T (*Figure 7—figure supplement 1*).

During Wave 2, Kilifi, Tana River, and Kwale observed their first major wave of infections. This wave started when most of the government COVID-19 restriction measures had been lowered or removed. For instance, international flights resumed on August 1, 2020, the operation of bars had resumed in September 2020, phased reopening of schools started in October 2020, and the curfew hours were moved to from 11 pm to 4 am. A total of 29 lineages were detected in Coastal Kenya during Wave 2, 9 of these had also been earlier detected during Wave 1.

Genomic data on GISAID database indicated that lineages B.1.530, B.1.549, and B.1.596.1 were predominantly detected in Kenya. The first sequenced cases of all these three lineages were identified in Taita Taveta County but the travel history of these individuals was indicated as 'unknown'. These lineages may have arisen in Kenya or another East Africa location that had limited genomic surveillance, for example, in Tanzania. Lineage B.1.530 has six characteristic mutations including spike P681H change adjacent to the biologically important furin cleavage site, lineage B.1.549 has seven characteristic mutations, five occurring in the ORF1a or ORF1b while lineage B.1.596.1 has eight lineage defining mutation 3 in ORF6 and three in N protein (*Figure 7—figure supplement 1*).

Three of the four Kenya-specific lineages were later observed in other countries albeit in small numbers. Lineage B.1.530 was detected in seven countries, namely, Germany (n = 3), the USA (n = 3), Rwanda (n = 1), Australia (n = 1), Japan (n = 1), and the Netherlands (n = 1). Lineage B.1.549 was detected in four countries, namely, England (n = 20), the USA (n = 4), Madagascar (n = 3), and Canada (n = 1). Lineage B.1.596.1 was detected in six countries, namely, the USA (n = 21), Sweden (n = 12), Australia (n = 2), Fiji (n = 1), Finland (n = 1), and India (n = 1). Note that the ancestral location state reconstruction analysis detected up to 105 virus exportation events from the Coastal Kenya counties to the rest of the world.

Lineage B.1.351 was first detected in Kilifi in November 2020 in a local with no travel history and later in two asymptomatic international travellers of South Africa nationality. Lineage B.1.1.7 was detected in a local who presented to a Mombasa clinic in the second week of January 2021 and in the subsequent weeks up to the end of the period covered by this analysis (February 2021), only one additional B.1.1.7 case was detected unlike lineage B.1.351, which continued to be detected sporadically in January and February 2021. Overall, only a minor increase in cases was observed in January–February 2021, despite the arrival of these VOCs before they subsequently resulted in the third national wave of infection recorded March–April 2021.

Despite the very large number of lineages detected globally (>900) during our study period, only a small fraction (n = 41, <5%) of these were documented in Coastal Kenya (*O'Toole et al., 2021*). Notably, two VOC lineages were already extensively spread across Eastern Africa (B.1.351), Africa (B.1.351), and worldwide (B.1.1.7) in the last quarter of 2020 unlike for Coastal Kenya. Thus, it is interesting that whereas in some countries (e.g. South Africa) the second wave appeared to be majorly driven by emergence of new variants, in Coastal Kenya, this may not have been the case. A lag was observed in the VOC large-scale spread in Coastal Kenya perhaps due to its remoteness and public health measures in place during the period.

Our study contributes to improved understanding on SARS-CoV-2 introduction and transmission patterns in sub-Saharan Africa countries (*Bugembe et al., 2020*; *Butera et al., 2021*; *Githinji et al., 2021*; *Mashe et al., 2021*; *Wilkinson et al., 2021*). This knowledge has potential to inform the application of future mitigation strategies especially in light of the growing evidence that SARS-CoV-2 will be endemic in human populations (*Planas et al., 2021*). Our analysis reveals lineage prevalence patterns and routes of entry into Coastal Kenya. New variants were frequently introduced via Mombasa County, thus surveillance in the city may provide an early warning system of new variant introductions into the region. We also provide evidence that the first two waves of infection in Coastal Kenya were not driven by VOCs, indicating the presence of other important factors impacting and driving SARS-CoV-2 waves of infection.

Sampling bias is a limitation as (1) sequenced and non-sequenced samples differed significantly in the demographic characteristics, (2) only a small proportion of confirmed cases (<10%) were sequenced, prioritizing samples with a Ct value of <30.0, (3) the MoH case identification protocols were repeatedly altered as the pandemic progressed (*Githinji et al., 2021*), and (4) sampling intensity across the six coastal counties due to accessibility differences. This may have skewed the observed

lineage and phylogenetic patterns. There was considerable missingness in metadata (e.g. travel history, nationality, *Table 2*), which made it hard to integrate genomic and epidemiological data in an analysis. Due to amplicon drop-off, some of the analysed genomes were incomplete impacting the overall phylogenetic signal.

The accuracy of the inferred patterns of virus movement into and from Coastal Kenya is dependent on both the representativeness of our sequenced samples for Coastal Kenya and the comprehensiveness of the comparison data from outside Coastal Kenya. Our sequenced sample was proportional to the number of positive cases reported in the respective Coastal Kenya counties. Also, we carefully selected comparison data to optimize chances of observing introductions occurring into the coastal region (e.g. by using all Africa data). But still there remained some important gaps, for example, non-coastal Kenya genomic data was limited (n = 605). Despite this, we think the results from ancestral state reconstruction indicate that Mombasa is a major gateway for variants entering Coastal Kenya is consistent with (1) the county showing the highest number lineages circulating during the study period compared to the other five remaining coastal counties Kenya, (2) approximately half of the detected lineages in Coastal Kenya had their first case identified in Mombasa, (3) Mombasa had an early wave of infections compared to the other coastal counties, and (4) Mombasa is the most well-connected county in the region to the rest of the world (large international seaport and airport and major railway terminus and several bus terminus).

In conclusion, we show that the first two SARS-CoV-2 waves in Coastal Kenya observed transmission of both newly introduced and potentially locally evolved lineages, many of them being non-VOCs. Approximately 50% of lineage introductions into the region occurred through Mombasa City. Our findings are consistent with mathematical modelling conclusion that it is more likely that relaxation or removal of some of the government COVID-19 countermeasures could have facilitated the second wave of SARS-CoV-2 infections in Kenya (*Brand et al., 2021*). Based on our observations of local distinctive phylogenies and the predominance of inter-county transmission, we suggest focusing COVID-19 control strategies on local transmission rather than international travel.

## Acknowledgements

We thank (1) the members of the six coastal counties of Kenya RRTs for collecting the samples analysed here; (2) the members of the COVID-19 KWTRP Testing Team who tirelessly analysed the samples received at KWTRP to identify positives (see full list of members below); (3) the KWTRP data entry team; (4) laboratories that have shared sequence data on GISAID that we included as comparison data in our analysis (see list in Supplementary file 7); and (5) the KRISP team in South Africa for sharing the scripts we used in the import/export analysis and AFRICA-CDC for facilitating Africa genomics training. This article was published with permission of the Director of KEMRI. *Members of COVID-19 Testing Team at KWTRP:* Agnes Mutiso, Alfred Mwanzu, Angela Karani, Bonface M Gichuki, Boniface Karia, Brian Bartilol, Brian Tawa, Calleb Odundo, Caroline Ngetsa, Clement Lewa, Daisy Mugo, David Amadi, David Ireri, Debra Riako, Domtila Kimani, Edwin Machanja, Elijah Gicheru, Elisha Omer, Faith Gambo, Horace Gumba, Isaac Musungu, James Chemweno, Janet Thoya, Jedida Mwacharo, John Gitonga, Johnstone Makale, Justine Getonto, Kelly Ominde, Kelvias Keter, Lydia Nyamako, Margaret Nunah, Martin Mutunga, Metrine Tendwa, Moses Mosobo, Nelson Ouma, Nicole Achieng, Patience Kiyuka, Perpetual Wanjiku, Peter Mwaura, Rita Warui, Robinson Cheruiyot, Salim Mwarumba, Shaban Mwangi, Shadrack Mutua, Sharon Owuor, Susan Njuguna, Victor Osoti, Wesley Cheruiyot, Wilfred Nyamu, Wilson Gumbi and Yiakon Sein. *Funding* This work was supported by the National Institute for Health and Care Research (NIHR) (project references 17/63/82 (PI JN) and 16/136/33 (PI MW)) using UK Aid from the UK Government to support global health research, The UK Foreign, Commonwealth and Development Office and Wellcome Trust (grant# 220985). The KEMRI-Wellcome Core award 203077/Z/16/Z from Wellcome award to PB supports the ongoing testing and thereafter the National COVID Testing Africa AAPs/Centre Wellcome Award, 222574/Z/21/Z, to PB and LIO-O supports the ongoing testing that identifies positive samples for sequencing. Members of COVID-19 Testing Team at KWTRP were supported by funding received by Dr Marta Maia (BOHEMIA study funded UNITAID), Dr Francis Ndungu (Senior Fellowship and Research and Innovation Action (RIA) grants from EDCTP), Dr Eunice Nduati (USAID grant to IAVI: AID-OAA-A-16–00032) and Prof. Anthony Scott (PCIVS grant from GAVI). MC and MVTP were supported by the Wellcome Trust and FCDO – Wellcome Epidemic Preparedness – Coronavirus (AFRICO19, grant agreement number 220977/Z/20/Z), from the MRC

(MC_UU_1201412) and from the UK Medical Research Council (MRC/UKRI) and FCDO (DIASEQCO, grant agreement number NC_PC_19060). SD acknowledges support from the *Fonds National de la Recherche Scientifique* (F.R.S.-FNRS, Belgium; grant no. F.4515.22), from the Research Foundation – Flanders (*Fonds voor Wetenschappelijk Onderzoek-Vlaanderen*, FWO, Belgium; grant no. G098321N), and from the European Union Horizon 2020 project MOOD (grant agreement no. 874850). The views expressed in this publication are those of the authors and not necessarily those of NIHR, the Department of Health and Social Care, Foreign Commonwealth and Development Office, Wellcome Trust, or the UK government.

## Additional information

### Funding

| Funder | Grant reference number | Author |
|---|---|---|
| National Institute for Health and Care Research | 17/63/82 | D James Nokes |
| National Institute for Health and Care Research | 16/136/33 | Charles N Agoti Samson Kinyanjui George Warimwe D James Nokes George Githinji |
| Wellcome Trust | 220985 | D James Nokes George Githinji |
| Wellcome Trust | 203077/Z/16/Z | Edwine Barasa Benjamin Tsofa Philip Bejon |
| Wellcome Trust | 220977/Z/20/Z | My Phan Matthew Cotten |
| Medical Research Council | MC_PC_20010 | My Phan Matthew Cotten |
| H2020 European Research Council | n°874850 | Simon Dellicour |

The funders had no role in study design, data collection and interpretation, or the decision to submit the work for publication. For the purpose of Open Access, the authors have applied a CC BY public copyright license to any Author Accepted Manuscript version arising from this submission.

### Author contributions

Charles N Agoti, Conceptualization, Data curation, Formal analysis, Methodology, Visualization, Writing – original draft, Writing – review and editing, Investigation; Lynette Isabella Ochola-Oyier, Ambrose Agweyu, Conceptualization, Investigation, Project administration, Resources, Supervision, Writing – review and editing; Simon Dellicour, Formal analysis, Visualization, Writing – review and editing; Khadija Said Mohammed, Leonard Ndwiga, Data curation, Formal analysis, Investigation, Methodology, Project administration, Writing – review and editing; Arnold W Lambisia, John M Morobe, Maureen W Mburu, Donwilliams O Omuoyo, Edidah M Ongera, Data curation, Formal analysis, Investigation, Methodology, Writing – review and editing; Zaydah R de Laurent, Data curation, Formal analysis, Investigation, Methodology, Project administration, Validation, Writing – review and editing; Eric Maitha, Thani Suleiman, Mohamed Mwakinangu, John K Nyambu, John Otieno, Barke Salim, Investigation, Methodology, Project administration, Supervision, Writing – review and editing; Benson Kitole, John N Kiiru, Investigation, Methodology, Project administration, Resources, Supervision; Jennifer Musyoki, Investigation, Project administration, Supervision, Writing – review and editing; Nickson Murunga, Data curation, Formal analysis, Investigation, Methodology, Validation, Writing – review and editing; Edward Otieno, Data curation, Formal analysis, Validation, Writing – review and editing; Kadondi Kasera, Data curation, Investigation, Methodology, Resources, Writing – review and editing; Patrick Amoth, Funding acquisition, Investigation, Project administration,

Resources, Supervision; Mercy Mwangangi, Conceptualization, Investigation, Methodology, Project administration, Resources, Supervision; Rashid Aman, Funding acquisition, Investigation, Methodology, Project administration, Resources, Supervision; Samson Kinyanjui, Conceptualization, Funding acquisition, Investigation, Project administration, Resources, Supervision, Writing – review and editing; George Warimwe, Data curation, Funding acquisition, Investigation, Methodology, Project administration, Resources, Writing – review and editing; My Phan, Formal analysis, Investigation, Methodology, Writing – review and editing; Matthew Cotten, Conceptualization, Funding acquisition, Project administration, Resources, Writing – review and editing; Edwine Barasa, Methodology, Project administration, Resources, Supervision, Writing – review and editing; Benjamin Tsofa, Conceptualization, Funding acquisition, Methodology, Project administration, Resources, Supervision, Writing – review and editing; D James Nokes, Conceptualization, Funding acquisition, Investigation, Project administration, Resources, Supervision, Writing – original draft, Writing – review and editing; Philip Bejon, Conceptualization, Formal analysis, Funding acquisition, Project administration, Resources, Supervision, Writing – original draft, Writing – review and editing; George Githinji, Conceptualization, Formal analysis, Funding acquisition, Investigation, Methodology, Project administration, Software, Supervision, Validation, Writing – review and editing

### Author ORCIDs
Charles N Agoti ⓘ http://orcid.org/0000-0002-2160-567X
Simon Dellicour ⓘ http://orcid.org/0000-0001-9558-1052
John M Morobe ⓘ http://orcid.org/0000-0003-2398-6717
Donwilliams O Omuoyo ⓘ http://orcid.org/0000-0003-3900-5354
Edward Otieno ⓘ http://orcid.org/0000-0002-8014-7306
My Phan ⓘ http://orcid.org/0000-0002-6905-8513
Matthew Cotten ⓘ http://orcid.org/0000-0002-3361-3351
Benjamin Tsofa ⓘ http://orcid.org/0000-0003-1000-1771
D James Nokes ⓘ http://orcid.org/0000-0001-5426-1984
George Githinji ⓘ http://orcid.org/0000-0001-9640-7371

### Ethics
Human subjects: Samples analysed here were collected under the Ministry of Health protocols as part of the national COVID-19 public health response. The whole genome sequencing study protocol was reviewed and approved by the Scientific and Ethics Review Committee (SERU) at Kenya Medical Research Institute (KEMRI), Nairobi, Kenya (SERU protocol #4035). Individual patient consent was not required by the committee for the use of these samples for studies of genomic epidemiology to inform public health response.

### Decision letter and Author response
Decision letter https://doi.org/10.7554/eLife.71703.sa1
Author response https://doi.org/10.7554/eLife.71703.sa2

## Additional files

### Supplementary files
• Supplementary file 1. Kenya government public health response and intervention to the COVID-19 pandemic.

• Supplementary file 2. Details on GISAID accession IDs, county of sampling, date of sample collection, assigned lineage, assigned clade, and the nucleotide sequences of the presented 1139 Coastal Kenya severe acute respiratory syndrome coronavirus 2 (SARS-CoV-2) genomes.

• Supplementary file 3. History of lineages detected in Coastal Kenya during the study period.

• Supplementary file 4. Patterns of Pango lineage detection at the various scales of observation analysed.

• Supplementary file 5. A summary of the top 10 detected Pango lineages detected in the different scales of observation investigated.

• Supplementary file 6. Summary output from separate runs of the import/export ancestral state reconstruction (ASR) analysis.

• Supplementary file 7. Acknowledgement of investigators and laboratories that have deposited

genomic data into GISAID database that we used to place the Coastal Kenya genomes into the global context.

• Transparent reporting form

• Reporting standard 1. STROBE Checklist.

• Source code 1. The R scripts used in the generation of the main text figures presented in the article.

## Data availability

(1) Sequence data have been deposited in GISAID database under accession numbers provided in Supplement File 2. (2) Source Data files have been provided for Figures 1-2 and 4-10. (3) Source Code associated with the figures has been uploaded (Source Code File 1) and also been made available through Harvard Dataverse.

The following dataset was generated:

| Author(s) | Year | Dataset title | Dataset URL | Database and Identifier |
|-----------|------|---------------|-------------|--------------------------|
| Agoti CN | 2021 | Replication Data for: Genomic surveillance reveals the spread patterns of SARS-CoV-2 in coastal Kenya during the first two waves | https://doi.org/10.7910/DVN/4ZZYIM | Harvard Dataverse, 10.7910/DVN/4ZZYIM |

The following previously published dataset was used:

| Author(s) | Year | Dataset title | Dataset URL | Database and Identifier |
|-----------|------|---------------|-------------|--------------------------|
| Githinji G | 2021 | Genomic epidemiology of SARS-CoV-2 in coastal Kenya (March - July 2020) | https://github.com/george-githinji/sars-cov-2-early-phase-manuscript | Github, 8402936 |

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
