## [Editor Report]

It is important to describe patterns of SARS-CoV-2 spread across the globe, beyond high-income countries. This study provides and evaluates SARS-CoV-2 sequence data from ~1200 PCR confirmed COVID-19 patients in Coastal Kenya to characterize phylogenetically likely importation and geographic infection routes, as well as the emergence of geographically distinct SARS-CoV-2 lineages.

---

## [Decision Letter]

**Decision letter after peer review:**

Thank you for submitting your article "Transmission networks of SARS-CoV-2 in coastal Kenya during the first two waves: a retrospective genomic study" for consideration by *eLife*. Your article has been reviewed by 2 peer reviewers, and the evaluation has been overseen by a Reviewing Editor and David Serwadda as the Senior Editor. The reviewers have opted to remain anonymous.

Essential revisions:

– You will see that both reviewers raised likely sampling bias as a major limitation of the study. A revision should include analyses that address this issue. One approach suggested would be to use the full African sequence data set and a balanced subsample of the non-African sequences.

– Authors should make an attempt to respond to all other issues raised by the reviewers.

*Reviewer #1 (Recommendations for the authors):*

– It’s quite a pain to review this text because figures are separate from figure legends, and not included at the most suitable position in the main text.

– I don t understand the flow chart, why 24% of PCR + samples sequenced

– line 151: please clarify what version of Pangolin was used

– line 168: given the limited sample sizes, reported percentages with one decimal digit convey a sense of precision that is not warranted. I suggest to remove decimal digits.

– line 172: I recommend to describe sample sizes as well.

– line 182: I did not follow this sentence. Please clarify.

– line 193: To interpret the biweekly distribution of lineages in each location, it is essential to consider sample sizes. How wide actually are Agresti Coull confidence intervals for each lineage in each biweek? Considering the data by 1 month or 2 month intervals may be more suitable, whilst still allowing for replacement dynamics to be recognised.

– line 199: Do you mean January 2021?

– line 225: can you date the SARS-CoV-2 phylogenies and estimate the time of introduction of B117 in your coastal sample, or perhaps more simply report the first date of diagnosis. This could provide indication on the time elapsed until VOCs arrived in coastal Kenya.

– line 231: the statement "Greater than 95% of the lineages comprising infections globally were not seen in the Coastal Kenya samples". Please specify the time frame, and support this statement in a Supplementary Figure or Table. It would also be interesting to report on the proportion of lineages in the Coastal Kenya infections that are not seen in non-Africa global samples, and non-Kenya samples.

– line 244: "with some clusters comprising genomes detected across multiple counties" -> these clusters are not well visualised in Figure 5A, and it would be of interest to visualise and discuss them in more detail. Are they all starting in Mombasa? If needed, Figure 5B-D could be moved to the Supplement. Addendum: having read on, I see that you discuss several phylogenetic subtrees in more detail. Are these all those seen in Figure 5A, or were they somehow selected?

– line 256: "possess significant genetic diversity consistent with wide scale spread" -> I am not sure how "significance" was determined, and why the level of diversity is consistent with wide scale spread.

– line 265: I am not sure what the analyses presented in this paragraph aim to show.

– line 280: can the data in Figure 8A presented in a different way so that numbers or proportions can be easily read off? As this figure stands, the only meaningful qualitative observation I can make is that Mombasa drives inter county spread.

– line 307: what was the purpose of the amino acid evolution analysis?

– line 375: this sentence is not clear to me.

– line 416: this sentence is not clear to me.

– line 419: it is not clear to me how exactly this study informs on optimising local interventions.

– line 520: requiring >80% coverage seems high. Why could >50% coverage not be used for this analysis, and would >50% coverage have resulted in more substantial representation of African sequences in the data sets?

*Reviewer #2 (Recommendations for the authors):*

As mentioned, this is a very well-written paper that presents several well-constructed analyses for identifying and tracking SARS-CoV-2 lineages across coastal Kenya. However, the conclusions presented aren't particularly surprising, and the paper is mostly descriptive. I think adding some discussion of restrictions data or other local context would make the paper a stronger fit for a journal such as *eLife*.

Small comments for consideration by the authors:

Figure 1A. This is a small point, but I didn't find the colors particularly intuitive here. Could the authors use lighter colors for fewer cases and darker colors for more cases, rather than the divergent color scheme currently used?

Line 99. Could the authors define the Oxford Stringency index briefly in the main text?

Line 151. I believe that the updated name for lineages assigned by the PANGOLIN software is "Pango lineages".

Figure 3A. Could the authors more clearly indicate which wave each month belongs to? It would also be useful to add some information about when each lineage was present globally, either here or in another figure panel (I know that this information is present in Table 2, but a visual representation would be helpful).

Figure 3C. Could the authors somehow visually indicate which identified lineages were unique to Kenya? Perhaps by cross-hatching those colored bars?

Line 217. I think it is better to describe a lineage as "predominantly found in X country" rather than to call it a "Rwandan lineage", for example.

Line 225. Could the authors be more consistent in their use of Pango lineages versus the WHO Α/Β/etc. nomenclature? Switching back and forth is a bit confusing, especially since the connection between the two nomenclatures isn't explicitly introduced.

Line 256. How can the reader evaluate the authors' claim of "significant diversity" by looking at a time tree? Could the authors provide the maximum likelihood trees for these lineages?

Line 259. I think the authors meant to say "implying multiple export (or import) events" ?

Figure 7. If possible, please avoid using red and green colors in the same figure, as these colors are hard to distinguish for many people.

Line 269. Why is it interesting that some lineages were more or less divergent from the Wuhan reference? Isn't this mostly a function of when the lineages emerged? Is there something worth noting on the root-to-tip plot shown in a previous figure?

Line 289. I think the authors mean ONT not OTN.

---

## [Author Response]

Essential revisions:– You will see that both reviewers raised likely sampling bias as a major limitation of the study. A revision should include analyses that address this issue. One approach suggested would be to use the full African sequence data set and a balanced subsample of the non-African sequences.

We thank the editor for an opportunity to resubmit a revised draft.

In the revised manuscript we have included the full African sequence data set that was available on GISAID as of November 2021, sampled during the study period and passing our set basic quality control filters (n=21,150) and a selected subsample for the rest of the world (n=21,093) – see Figure 3—figure supplement 1. The latter was randomly identified using an in-house R code that forced spatial-temporal representation. Basically, a maximum of 30 genomes were accepted per country per month per year (page 36, lines 886-893). We ended up with a select 42,243 genomes and all these were utilized in the widening scales of observation lineage tracking analysis.

However, for phylogenetic comparisons, we had to undertake subsampling of the comparison global genomes due to the large computational requirements to run a maximum likelihood phylogeny with >40,000 taxa. Our subsampling strategy was to limit the inclusion of global genomes to those Pango lineages identified in Coastal Kenya. We then split the resulting dataset into two groups (each ~9500 taxa) to make analysis tractable, see details in Figure 3—figure supplement 1.

During the review period, an additional 389 genomes from Coastal Kenya became available and these have been added into this revised analysis.

We believe this computational subsampling from all available global and African genomes sequences is perhaps the best argument against bias in our observations.

– Authors should make an attempt to respond to all other issues raised by the reviewers.

We have carefully responded to all comments and concerns raised by the reviewers, see below.

Reviewer #1 (Recommendations for the authors):– It’s quite a pain to review this text because figures are separate from figure legends, and not included at the most suitable position in the main text.

We apologize for that experience. In the resubmitted manuscript, we have fully followed the editorial guidelines on how to organize the manuscript at resubmission stage, we hope this improves readability.

– I don t understand the flow chart, why 24% of PCR + samples sequenced

We thank the reviewer for spotting this, there was an error here in the previous version of the manuscript which we have now corrected. In the revised manuscript, we have clarified that we sequenced 18% of the RT-PCR positives (i.e. 1139/6329).

– line 151: please clarify what version of Pangolin was used

This has been clarified (page 6: line 226-227, Pangolin version 3.1.16).

– line 168: given the limited sample sizes, reported percentages with one decimal digit convey a sense of precision that is not warranted. I suggest to remove decimal digits.

We concur and have removed the decimal digits.

– line 172: I recommend to describe sample sizes as well.

This is a good suggestion, and this is now included in Page 11 lines 515-526 and page 12, lines 553-564.

– line 182: I did not follow this sentence. Please clarify.

Apologies for the confusing sentence. The sentence has now been removed as the said lineage has been reclassified by updated Pango lineage assignment.

– line 193: To interpret the biweekly distribution of lineages in each location, it is essential to consider sample sizes. How wide actually are Agresti Coull confidence intervals for each lineage in each biweek? Considering the data by 1 month or 2 month intervals may be more suitable, whilst still allowing for replacement dynamics to be recognised.

In the revised draft, we have aggregated the data by month. To show the sample sizes, Figure 4C now presents the number of genomes per month per county rather than proportions. The version of the figure showing proportions has been moved to the supplementary section (Figure 4—figure supplement 2)

We agree that presenting Agresti Coull confidence intervals for each lineage in each biweek is a robust way of evaluating significant temporal changes. However, with many of the Pango lineages occurring in small numbers, deriving such confidence intervals is challenging and will be difficult to interpret.

– line 199: Do you mean January 2021?

Yes, thank you for spotting this, the date has been corrected.

– line 225: can you date the SARS-CoV-2 phylogenies and estimate the time of introduction of B117 in your coastal sample, or perhaps more simply report the first date of diagnosis. This could provide indication on the time elapsed until VOCs arrived in coastal Kenya.

We thank the reviewer for this suggestion. The first date of diagnosis for both B.1.1.7 and B.1.351 in our coastal samples has now been provided in lines page 12, lines 545-546. “Β (lineage B.1.351, n=26, first detected in a Kilifi sample collected on 4th November 2020); and Α lineage B.1.1.7, n=2, first detected on 14^th^ January 2021 in a Mombasa sample.”

– line 231: the statement "Greater than 95% of the lineages comprising infections globally were not seen in the Coastal Kenya samples". Please specify the time frame, and support this statement in a Supplementary Figure or Table. It would also be interesting to report on the proportion of lineages in the Coastal Kenya infections that are not seen in non-Africa global samples, and non-Kenya samples.

The timeframe is 1^st^ March 2020 to 28^th^ February 2021, and this is now clarified in the revised manuscript (page 13 line 612-617) now reads as below.

“Greater than 95% (909/950) of the lineages comprising infections globally for the analyzed sub-sample collected between 1st March 2020 and 28th February 2021 were not seen in the Coastal Kenya samples, Supplementary File 3. Only two lineages were observed in the Coastal Kenya sample but were not in the global subsample; lineage N.8 (predominantly found in Kenya) and lineage B.1.593 (predominantly found in USA), Figure 3.”

We have added a table in the supplementary material (Supplementary File 3) showing the Pango lineage overlap between the different scales of observation.

– line 244: "with some clusters comprising genomes detected across multiple counties" -> these clusters are not well visualised in Figure 5A, and it would be of interest to visualise and discuss them in more detail. Are they all starting in Mombasa? If needed, Figure 5B-D could be moved to the Supplement. Addendum: having read on, I see that you discuss several phylogenetic subtrees in more detail. Are these all those seen in Figure 5A, or were they somehow selected?

Considering the reviewer’s suggestion, previous Figure 5B has now been moved to the supplementary material (currently Figure 8—figure supplement 2) while previous 5C and D are dropped from the manuscript since their key message is no longer core to the manuscript’s main storyline.

In the revised manuscript, Figure 9 provides a zoomed in view of the top 9 lineages identified in the Coastal Kenya (A, B, B.1, B.1.1, B.1.351, B.1.530, B.1.549, B.1.596.1 and N.8). Equivalent phylogenies that are time-resolved are provided in Figure 9 —figure supplement 1.

– line 256: "possess significant genetic diversity consistent with wide scale spread" -> I am not sure how "significance" was determined, and why the level of diversity is consistent with wide scale spread.

We have rephrased to “considerable within lineage diversity (highest in the predominant lineage B.1), this consistent with ongoing within lineage SARS-CoV-2 genetic evolution”, page 24, lines 665-667.

– line 265: I am not sure what the analyses presented in this paragraph aim to show.

We agree that the said analysis is not core to the manuscript thus we have deleted it from the revised manuscript.

– line 280: can the data in Figure 8A presented in a different way so that numbers or proportions can be easily read off? As this figure stands, the only meaningful qualitative observation I can make is that Mombasa drives inter county spread.

Thanks for this suggestion. We have revised the plot to show not only the patterns of intercounty transmission but also the similarities and differences in patterns between wave one and wave two.

The observation that Mombasa was the main driver of inter-county spread is indeed the main conclusion from this analysis and we hope that the revised Figure 10A makes this clearer. The aggregated number of imports into each county is provided in Table 4.

In Figure 10 panel B we compare the total location transition events between the Wave one period and the Wave two period, and the observed numbers can be easily read from these bar charts.

We have added Figure 10C that shows the temporal trends of imports, exports and inter-county location transition events during the study period clearly showing the change from mostly imports during Wave one to mostly intercounty transmission during Wave two.

In Figure 10 panel D, we provide bar plots to show the quantitative distribution of the estimated origins of imports into each county.

– line 307: what was the purpose of the amino acid evolution analysis?

We agree that the amino acid changes analysis is not core to the current manuscript and have dropped it from the revised manuscript.

– line 375: this sentence is not clear to me.

Apologies for this confusing sentence, we have rephrased the sentence to “We note from the widening scales of observation analysis that despite the very large number of lineages detected globally (>900) during our study period (1st March 2020 to 28th February 2021, only a small fraction (n=41, <5%) of these were documented in Coastal Kenya (O’Toole et al., 2021)).” See Page 18 line 860-861.

– line 416: this sentence is not clear to me.

Thanks for alerting us. We have rephrased the sentence to:

“Thus, the current analysis is consistent with findings from mathematical models that it is more likely that the relaxation or removal of some of the government COVID-19 countermeasures (reopening of international airspace in August 2020, reopening of bars and restaurants in September 2020, partial reopening learning institutions in October 2020) was associated with a return to pre-pandemic mobility levels among the higher social economic group that facilitated the second wave of SARS-CoV-2 infections in Kenya including for the coastal region (Brand et al., 2021).” See Page 20, line 932-938.

– line 419: it is not clear to me how exactly this study informs on optimising local interventions.

We have added the following paragraph to the discussion to clarify this point.

“Improved understanding of SARS-CoV-2 lineage introductions and local spread during the early waves has potential to inform the application of future mitigation strategies. There is growing evidence that SARS-CoV-2 will be endemic in human populations for the foreseeable future (Planas et al., 2021). Our analysis reveals lineage prevalence patterns and routes of entry into Coastal Kenya that are likely to persist. Our finding that during the early period, lineages in the first two infection waves in coastal Kenya were frequently introduced via Mombasa County, support strengthening surveillance in Mombasa for an early warning system of new variant introductions into the region. We also provide evidence that unlike the SARS-CoV-2 wave numbers 3-5 in Kenya, the early SARS-CoV-2 waves (number 1 and 2) in Coastal Kenya were not driven by occurrence of a VOC indicating presence of other important factors in observed SARS-CoV-2 waves of infection” Page 19, lines 876-887.”

– line 520: requiring >80% coverage seems high. Why could >50% coverage not be used for this analysis, and would >50% coverage have resulted in more substantial representation of African sequences in the data sets?

The 80% cut-off is supported by:

a) PANGO lineages can only be assigned confidently by the pangolin toolkit when genome coverage of >70%.

b) Since lineage assignment is one of our key analyses and there is considerably large, good quality genomic data (>20,000 with >80% coverage) that is publicly available and representative across the Africa, we did not find it informative to include partial genomes from Africa (<80% coverage) that may also be unassignable to PANGO lineages.

c) The missing 20% (or 50%) in many of the GISAID genomes tends to be in the spike region which is phylogenetically the most informative part of the genome, so actually using even 80% genomes (rather than a higher percentage cutoff) is problematic because you miss a lot of informative sites. Dropping to 50% is probably not a good way to improve the analysis. Yes, you may increase the number of genomes, but they are missing the informative nucleotide changes.

d) In the revised analysis we were limited by computational power for the number of genome sequences included, hence did not wish to include more than the African genomes data set available on GISAID with > 80% coverage (n=21, 150), see Figure 3—figure supplement 1.

Reviewer #2 (Recommendations for the authors):As mentioned, this is a very well-written paper that presents several well-constructed analyses for identifying and tracking SARS-CoV-2 lineages across coastal Kenya. However, the conclusions presented aren't particularly surprising, and the paper is mostly descriptive. I think adding some discussion of restrictions data or other local context would make the paper a stronger fit for a journal such as eLife.

We thank the reviewer for the kind words. We have added a sub analysis and comments on the restrictions, nationality, and travel data on page 10 lines 385-396 and page 10 lines 467479. Some of the results from this analysis are provided in Figure 7 and Figure 5—figure supplement 1 and 2.

Small comments for consideration by the authors:Figure 1A. This is a small point, but I didn't find the colors particularly intuitive here. Could the authors use lighter colors for fewer cases and darker colors for more cases, rather than the divergent color scheme currently used?

We have revised the Figure 1A as suggested. We hope that this improves the resolution and clarity of the figure.

Line 99. Could the authors define the Oxford Stringency index briefly in the main text?

The Oxford Stringency index is now described in the methods section of the manuscript. Page 8 Lines: 340-350,

“Oxford SI is based on nine response indicators rescaled to values of 0-100, with 100 being strictest (Hale et al., Noam Angrist). The nine response indicators used to form the SI are (a) school closures, (b) workplace closures, (c) cancellation of public events, (d) restrictions on public gatherings, (e) closures of public transport, (f) stay-at-home requirements, (g) public information campaigns, (h) restrictions on internal movements and (i) International travel controls.**”**

Line 151. I believe that the updated name for lineages assigned by the PANGOLIN software is "Pango lineages".

We thanks the reviewer for this correction, and we have updated the name to Pango lineages throughout the manuscript.

Figure 3A. Could the authors more clearly indicate which wave each month belongs to? It would also be useful to add some information about when each lineage was present globally, either here or in another figure panel (I know that this information is present in Table 2, but a visual representation would be helpful).

This is a useful suggestion, in the updated Figure 4A, we have added a vertical dashed line demarcates the date of transition from Wave one to Wave two.

Also in the revised manuscript, we have included a new Figure 3 which details the global context of the 43 lineages identified in Coastal Kenya.

Figure 3C. Could the authors somehow visually indicate which identified lineages were unique to Kenya? Perhaps by cross-hatching those colored bars?

Thanks for this useful suggestion. We have indicated the Kenya unique lineages in Figure 4B where we have stratified the detected lineages into those that were Kenya specific and those that were predominantly detected elsewhere (international lineages).

Line 217. I think it is better to describe a lineage as "predominantly found in X country" rather than to call it a "Rwandan lineage", for example.

Thanks, we agree, and we have corrected this in the revised manuscript.

Line 225. Could the authors be more consistent in their use of Pango lineages versus the WHO Α/Β/etc. nomenclature? Switching back and forth is a bit confusing, especially since the connection between the two nomenclatures isn't explicitly introduced.

This is a very good point, and we apologize for our inconsistent use of the nomenclature. This has been corrected in the revised manuscript where we have used the Pango lineage nomenclature throughout. On first use of the Pango lineage and at key points in the manuscript (e.g. Table headings, figures) we have added the WHO nomenclature in parentheses. We hope that this improves the readability and limits confusion.

Line 256. How can the reader evaluate the authors' claim of "significant diversity" by looking at a time tree? Could the authors provide the maximum likelihood trees for these lineages?

This is a valid concern. In the revised manuscript, we have provided mutation-resolved phylogenetic trees for the specific lineages (Figure 9). The time-resolved lineage-specific phylogenetic trees have been moved to the supplementary material section (Figure 9—figure supplement 1).

Line 259. I think the authors meant to say "implying multiple export (or import) events" ?

Yes, thanks, that has been corrected.

Figure 7. If possible, please avoid using red and green colors in the same figure, as these colors are hard to distinguish for many people.

Thanks for pointing this out. In the revised manuscript, we dropped the previous Figure 7 with this mixed red/green. We have also checked all other figures for appropriate color use.

Line 269. Why is it interesting that some lineages were more or less divergent from the Wuhan reference? Isn't this mostly a function of when the lineages emerged? Is there something worth noting on the root-to-tip plot shown in a previous figure?

Considering our focus in this manuscript is the SARS-CoV-2 epidemiology lineage spatial-temporal dynamics rather than their molecular evolution, the analysis considering divergence within the lineages has been dropped from the revised manuscript.

Line 289. I think the authors mean ONT not OTN.

We thank the reviewer for noting this, it has been corrected.

References

Brand, S.P.C., Ojal, J., Aziza, R., Were, V., Okiro, E.A., Kombe, I.K., Mburu, C., Ogero, M., Agweyu, A., Warimwe, G.M., Nyagwange, J., Karanja, H., Gitonga, J.N., Mugo, D., Uyoga, S., Adetifa, I.M.O., Scott, J.A.G., Otieno, E., Murunga, N., Otiende, M., Ochola-Oyier, L.I., Agoti, C.N., Githinji, G., Kasera, K., Amoth, P., Mwangangi, M., Aman, R., Ng'ang'a, W., Tsofa, B., Bejon, P., Keeling, M.J., Nokes, D.J., Barasa, E., 2021. COVID-19 transmission dynamics underlying epidemic waves in Kenya. Science 374(6570), 989-994.

Githinji, G., de Laurent, Z.R., Mohammed, K.S., Omuoyo, D.O., Macharia, P.M., Morobe, J.M., Otieno, E., Kinyanjui, S.M., Agweyu, A., Maitha, E., Kitole, B., Suleiman, T., Mwakinangu, M., Nyambu, J., Otieno, J., Salim, B., Kasera, K., Kiiru, J., Aman, R., Barasa, E., Warimwe, G., Bejon, P., Tsofa, B., Ochola-Oyier, L.I., Nokes, D.J., Agoti, C.N., 2021. Tracking the introduction and spread of SARS-CoV-2 in coastal Kenya. Nature Communications 12(1), 4809.

O’Toole, Á., Scher, E., Underwood, A., Jackson, B., Hill, V., McCrone, J.T., Colquhoun, R., Ruis, C., Abu-Dahab, K., Taylor, B., Yeats, C., du Plessis, L., Maloney, D., Medd, N., Attwood, S.W., Aanensen, D.M., Holmes, E.C., Pybus, O.G., Rambaut, A., 2021. Assignment of epidemiological lineages in an emerging pandemic using the pangolin tool. Virus Evolution.

Planas, D., Saunders, N., Maes, P., Guivel-Benhassine, F., Planchais, C., Buchrieser, J., Bolland, W.-H., Porrot, F., Staropoli, I., Lemoine, F., Péré, H., Veyer, D., Puech, J., Rodary, J., Baele, G., Dellicour, S., Raymenants, J., Gorissen, S., Geenen, C., Vanmechelen, B., Wawina -Bokalanga, T., Martí-Carreras, J., Cuypers, L., Sève, A., Hocqueloux, L., Prazuck, T., Rey, F., Simon-Loriere, E., Bruel, T., Mouquet, H., André, E., Schwartz, O., 2021. Considerable escape of SARS-CoV-2 Omicron to antibody neutralization. Nature.